# Lignocellulosic Biomass for the Fabrication of Triboelectric Nano-Generators (TENGs)—A Review

**DOI:** 10.3390/ijms242115784

**Published:** 2023-10-30

**Authors:** Omar P. Troncoso, Jim I. Corman-Hijar, Fernando G. Torres

**Affiliations:** Department of Mechanical Engineering, Pontificia Universidad Católica del Perú, Av. Universitaria 1801, Lima 15088, Peru; troncoso.op@pucp.pe (O.P.T.); jcormanh@pucp.edu.pe (J.I.C.-H.)

**Keywords:** triboelectric nanogenerators, TENGs, lignocellulose, self-powered devices, sensors, waste materials

## Abstract

Growth in population and increased environmental awareness demand the emergence of new energy sources with low environmental impact. Lignocellulosic biomass is mainly composed of cellulose, lignin, and hemicellulose. These materials have been used in the energy industry for the production of biofuels as an eco-friendly alternative to fossil fuels. However, their use in the fabrication of small electronic devices is still under development. Lignocellulose-based triboelectric nanogenerators (LC-TENGs) have emerged as an eco-friendly alternative to conventional batteries, which are mainly composed of harmful and non-degradable materials. These LC-TENGs use lignocellulose-based components, which serve as electrodes or triboelectric active materials. These materials can be derived from bulk materials such as wood, seeds, or leaves, or they can be derived from waste materials from the timber industry, agriculture, or recycled urban materials. LC-TENG devices represent an eco-friendly, low-cost, and effective mechanism for harvesting environmental mechanical energy to generate electricity, enabling the development of self-powered devices and sensors. In this study, a comprehensive review of lignocellulosic-based materials was conducted to highlight their use as both electrodes and triboelectric active surfaces in the development of novel eco-friendly triboelectric nano-generators (LC-TENGs). The composition of lignocellulose and the classification and applications of LC-TENGs are discussed.

## 1. Introduction

Lignocellulosic biomass is the most abundant organic and renewable material on Earth. It is generated by nature via photosynthetic chemical reactions [1]. The use of lignocellulose as an energy source dates back to the early stages of civilization. In this stage, wood burning was used to boil food and eliminate pathogens, thus improving human health. Due to the growing environmental awareness surrounding global warming, lignocellulose has been utilized as a source for producing biofuels. These biofuels currently account for approximately 14% of the world’s total primary energy supply and have a significantly lower environmental impact compared to fossil fuels [2]. The annual production of this lignocellulose biomass has been estimated at ~10^11^ tons in the last years [3]. However, the majority of the residues from this industry end up being discarded or burned [4]. Novel and efficient processing technologies must be developed to convert these lignocellulose residues into new materials or devices that can accelerate the necessary transition towards sustainability [5].

Lignocellulosic resources include agricultural residues, forest residues, energy crops, food wastes, and municipal and industrial waste [6]. The main components of lignocellulosic biomass are cellulose, lignin, and hemicellulose. The content of each of these components depends on the source. For instance, studies published elsewhere on agricultural waste report that wheat straw contains 37–41% cellulose, 27–32% hemicellulose, and 13–15% lignin [7]. In energy crops, empty fruit bunches and switchgrass contain 41–45% cellulose, 24–31% hemicellulose, and 12–21% lignin [8]. In forestry waste, hardwood and softwood stems contain 40–55% more cellulose than leaves, which only have a cellulose content of 15–20% [8]. Leaves have the highest hemicellulose content, ranging from 80 to 85%, compared to other lignocellulosic resources. In industrial waste, wastepaper contains approximately 60–70% cellulose [6].

The use of biofuels as an environmentally friendly energy source is primarily limited to large machinery. For small electronic devices, a different solution is needed. Triboelectric nanogenerators (TENGs) were first developed by Wang et al. [9] in 2012. They used the term “nanogenerator” to emphasize that a small electronic chip has the ability to harvest electrical energy from various mechanical energy sources, such as human motion during activities such as walking, breathing, and jogging. The dimensions of TENGs range from a few millimeters to centimeters, depending on the specific design and application [10].

Small electronic devices still use conventional batteries or rechargeable batteries, which are mainly composed of materials that are harmful, toxic, and dangerous to human health and the environment. TENGs could be a viable solution to the issue of using and disposing of numerous batteries over the lifespan of devices. TENGs are based on the triboelectric and electrification effects of materials. When two different materials, such as two dissimilar dielectric materials, come into contact, electrons migrate from one material to the other. This migration of electrons helps to balance the electrical potential between the connected materials and facilitates the exchange of charges. When these surfaces are separated, they become positively or negatively charged. When two materials are continuously in contact and then separated, electricity is generated due to the collision between them. This collision results in a high flow of electrons across the materials, creating and maintaining a potential difference.

These TENGs harvest low-frequency mechanical energy from the environment to generate electrical energy. The surfaces that store the electrical charges are called active surfaces. The electrical performance of TENGs is mainly based on the selection of the triboelectric materials of the active surfaces. These active surfaces are commonly made of synthetic materials such as PET, Kapton, Teflon, and nylon, among others. These materials are not recyclable and might produce a negative environmental impact. The use of renewable, biocompatible, and biodegradable materials in the development of TENGs addresses these disadvantages. In an earlier paper, Torres et al. [11] reviewed the use of various biopolymers in the development of novel TENGs. They found that the main challenges in Bio-TENGs are the low triboelectric properties and poor mechanical strength of the biopolymeric active surfaces that store the static charge. Several biopolymers, such as collagen and starch, have poor mechanical properties. Lignocellulosic materials, on the other hand, exhibit good mechanical properties. Lignocellulose-based TENGs (LC-TENGs) take advantage of the chemical composition and structure of lignocellulose to develop active materials. These materials can be used as electrodes or triboelectric layers. Lignocellulose-based materials are commonly composed of cellulose, lignin, and hemicellulose.

In this study, a comprehensive examination of lignocellulose, an emerging triboelectric material, was conducted. The composition of lignocellulose and the classification and applications of lignocellulose-based TENGs are discussed. Up to now, there has been no literature that explores the utilization of lignocellulose-based materials in the fabrication of triboelectric nanogenerators. This work addresses the extraction of refined and non-refined lignocellulose materials from various sources, including wood, seeds, leaves, and residues from the timber industry, agriculture, and urban recyclable materials.

## 2. Lignocellulosic Biomass: Components and Composition

A fundamental understanding of the individual components present in lignocellulosic biomass is necessary to comprehend its ability to bind with molecules and its inherent properties. Lignocellulosic biomass is primarily composed of cellulose (35–50%), hemicellulose (20–35%), and lignin (5–30%) as its three main constituents [12] (Figure 1a).

The inherent biodegradability and biocompatibility of lignocellulosic biomass and its derivatives make them promising alternatives in the fabrication of devices, such as electrical storage devices. They can be strong candidates for replacing fossil-based materials used in the energy industry and contribute to the urgent need to transition to a more environmentally friendly society [1,5]. In order to meet the demand for more sustainable energy and self-powered devices, research has focused on designing new generations of electric nanogenerators. TENGs can utilize bio-based renewable organic materials for the fabrication of both the active triboelectric surfaces and the encasement of the device. Among these nanogenerators, the triboelectric nanogenerator (TENG) is one of the most remarkable devices and can harvest various forms of mechanical energy found in nature or at human-interactive interfaces. It converts this energy into electrical energy with minimal environmental impact [1]. TENGs are based on the triboelectric effect and the electrostatic induction of materials. They require smart designs for various methods of inducing and enhancing surface contact electrification [1].

Cellulose has been extensively studied as a raw material for the development of new technological products in electronic and electrical devices. This is due to its abundance, low production costs, customizable chemical functionality, adjustable physical and mechanical properties, and inherent dielectric properties. The structure of cellulose (Figure 1b) is composed of abundant polar hydroxyl groups (-OH), which are beneficial for electrical applications due to dipole polarization. The large number of -OH groups increases the dielectric constant of cellulose-based materials. It also causes the formation of intermolecular hydrogen bonds, which decreases the free movement of dipoles and restrains dipole relaxation [13,14]. Cellulose fibers consist of crystalline and amorphous regions. Most of the hydrogen bonds are found in the crystalline region. The polarization in this region mainly includes electron polarization and atomic polarization. In the amorphous region, dipole polarization caused by the freedom of movement of polar groups, in addition to other polarizations, is possible [14].

Lignin represents 15–40% of the volume of plants [15,16]. It is a complex, three-dimensional molecular structure, made up of cross-linked polymers [6], which provides the structural integrity of plants [17]. These polymers are composed of three types of phenols. These phenols are coniferyl, sinapyl, and p-coumaryl alcohols (Figure 2a), which are cross-linked via enzymatic polymerization, resulting in a wide range of functional groups and linkages [18]. These bonds represent the primary obstacle in biomass separation processes. Lignin serves as a protective barrier for plant permeability and provides resistance against microbial attacks [6]. It is insoluble in water and acts as an adhesive, connecting the cellulose and hemicellulose [18]. Lignin has been treated as an undesirable residue after pretreatment to extract cellulose. For this reason, approximately 5 × 10^7^ tons of lignin are primarily disposed of by burning each year [19,20,21]. This scenario has been changing in recent years due to increasing environmental awareness, which promotes the use of renewable materials in various fields. The applications of lignin include the fabrication of emulsifiers, dyes, synthetic floorings, insulation, binding, thermosets, dispersal agents, and paints, as well as other applications in the electrochemical and medical/pharmaceutical fields [18,22,23,24,25].

Hemicellulose-coated cellulose fibrils are formed via the adhesion of short lateral chains composed of various sugars, including pentoses, hexoses, and acetylated sugars [6]. Figure 2b shows the main structures of xylan and glucomannan. These structures have special physical and chemical properties, being rich in hydroxyl groups and having a high molecular weight. Hemicellulose is nontoxic, nonionic, and biodegradable. Therefore, it has been used in the bio-packaging, food, pharmaceutical, medicine, cosmetic, textile, and papermaking industries [27].

## 3. Treatments Used to Isolate the Components of Lignocellulose Biomass

Pretreatment is necessary to overcome the recalcitrant nature of lignocellulose before it can be converted into other biomaterials. The term “recalcitrance” refers to the resistance of plant cell walls to deconstruction. Some factors contribute to high recalcitrance. These factors include a high crystalline structure of cellulose, a high lignin content that protects the cellulose, cellulose sheathing with hemicellulose, high fiber strength, and a low accessible surface area of cellulose [6,23]. Pretreatment processes involve disrupting the complex hierarchical structure (Figure 3a) of lignocellulosic biomass, allowing for the separation of its components [4,23]. Traditional pretreatment techniques involve biological, chemical, physical, or physicochemical approaches [23,24] (Figure 3b). These treatments usually involve the use of toxic and harmful chemical substances or require a high level of energy and expensive materials [4]. In recent years, green approaches have been developed to reduce the environmental impact of traditional pretreatment methods. These green approaches involve the use of deep eutectic solvents, microbes, supercritical fluids, and the steam explosion technique. However, these green methods usually have low yields and high production costs, and sometimes imply a long pretreatment time (several weeks) [4]. To overcome these disadvantages, researchers have been investigating new emerging technologies. These technologies use microwaves, ultrasound, gamma rays, electron beams, pulsed-electric fields, high hydrostatic pressure, and high-pressure homogenization [4].

Microwave irradiation features fast and uniform heat transfer, short reaction times, and low formation of by-products compared to conventional heating [30]. Ultrasound creates small, unstable cavitation bubbles in the lignocellulose structure. This treatment attacks the integrity of cell walls by cleaving the ether linkages between lignin and hemicelluloses, thereby increasing their extractability [4]. This method is also used to remove 90% of hemicellulose and lignin from lignocellulose waste materials, such as sugarcane bagasse [31]. Liquefaction of lignocellulosic materials is a crucial step in polymer synthesis and the production of environmentally friendly polymeric products. For example, polyurethane foams can be produced by liquefying wheat straw, wood laminating adhesive can be made from liquefied kenaf core, and wood particleboard can be produced by synthesizing polyester from liquefied wood [32,33,34,35]. Ultrasound techniques have also been used to assist with and enhance the performance of common lignocellulose liquefaction techniques. This helps to reduce the time and by-products involved while enabling the production of valuable chemicals from forest wood waste [36]. Sometimes, the synergistic effect of different approaches enables the isolation of components. Xie et al. [29] isolated cellulose nanofibers from bamboo using a combination of microwave liquefaction, chemical treatments, and ultrasonic treatments, resulting in high-purity nanofibers (Figure 3c). In spite of the fact that these new technologies, such as microwave- and ultrasound-based technologies, have aroused the interest of the scientific community, their industrial application has not yet been fully exploited, and new advances in R&D developments are still needed.

### 3.1. Extraction of Cellulose and Derivatives

Cellulose is obtained after removing hemicellulose, lignin, and other impurities (Figure 3a). Pure cellulose can be used to extract various types of nanocelluloses, including cellulose nanofibrils (CNF) and cellulose nanocrystals (CNC), via chemical, mechanical, and enzymatic treatments [37,38,39,40,41,42]. The yield of nanocellulose production varies depending on the source and processes used, ranging between 30% and 70% [43]. Figure 4a shows some of the treatments that can follow either a bottom-up or a top-down approach. Nanocellulose possesses a large surface area, distinctive optical properties, high crystallinity, and stiffness, as well as inherent biodegradability and renewability [6]. Nanocellulose has a wide range of applications in the biomedical, food, and energy fields (Figure 4b) [44,45,46,47]. There are bacteria that can synthesize nanocellulose fibers. This nanocellulose is known as bacterial cellulose (BC) and has been used for the fabrication of robust nanocellulose-based materials for various applications, including the development of innovative electronic devices [48,49,50,51].

### 3.2. Extraction of Lignin

Different types of solvents have been used to extract lignin from natural biomass. Ionic liquids (ILs) have been used as environmentally friendly solvents for extracting lignin. The lignin extraction yields reported using ILs ranged from 17 to 84% [52]. ILs possess interesting properties such as non-flammability, recyclability, non-volatility, low vapor pressure, and high boiling point [44,45]. Deep eutectic solvents (DESs) have also been used as an alternative green solvent for biomass pretreatment and lignin extraction. The extraction yields with DESs are between 20–75% [53]. DESs represent low toxicity and adequate biodegradability and are more cost-effective in the synthesis process than ILs [21]. Cronin et al. [46] used a DES composed of ChCl/LA to extract lignin from the hydrolysis and fermentation residue of corn stover hydrolysate. The obtained lignin showed a high purity at 94.7%, with a yield of 75%. Malaeke et al. [47] used a combination of DES pretreatment and ultrasonication to dissolve 50% of lignin. Watkins et al. [18] used a two-step technique composed of formic acid treatment followed by peroxy-formic acid treatment to extract lignin from jute, hemp, cotton, and wood pulp. Due to the high cost and long processing time associated with the ionic liquid solvent technique, Guiao et al. [54] proposed the use of a mechano-chemical processing method with 1-butyl-3-methylimidazolium chloride (BmimCl), followed by chemical treatment, to extract lignin from mixed hardwood flour (MHF). This technique achieved a maximum yield of 36.6% at a solid loading of 50% and a residence time of 45 min.

### 3.3. Extraction of Hemicellulose

There are many methods of hemicellulose extraction [27] that have been used and which do not have a significant environmental impact. These methods include hydrothermal extraction [55,56] and steam explosion [57,58]. Each treatment shows an extraction rate (ER) of 40–80% and 50–80%, respectively. Meanwhile, alkaline extraction [59,60,61] (extraction rate: 50–90%) and acid extraction [62] (ER: 70–85%), as well as organosolv fractionation [63] (ER: 20–97%), demonstrate high extraction rates and purity. However, these methods can have a significant environmental impact. Chemical treatments that have a low environmental impact include IL [64,65] (ER: 20–80%) and DES extraction [66,67] (ER: 40–70%).

## 4. TENGs Based on Lignocellulose and Its Derivatives

As described in a previous section, lignocellulosic materials can be used as electroactive materials or substrates in TENGs due to their chemical structure and properties. Because of the large number of polar molecules, bonds, and functional groups (such as OH), cellulose-based materials can be used as triboelectric positive materials. This material can be carbonized to achieve a carbon structure with higher conductivity than pure cellulose material, enabling it to serve as an electrode. Lignocellulose-derived materials can also serve as a matrix for the insertion of nanoparticles, thereby enhancing dielectric properties or conductivity. This makes them suitable for use as triboelectric materials or electrodes. TENGs are based on triboelectrification and the electrostatic effect. The displacement current generated via the contact or friction between two materials is collected by the electrodes. This system can be easily explained in terms of the capacitive model, which integrates three capacitors in a series. This integration occurs due to the interaction between triboelectric materials and the air space between electrodes (Figure 5a). Two materials need to be in contact or rubbed together to achieve polarization, which induces a change in charge in the electrodes. This change allows the flow of electrons through the load that is connected between them. In Figure 5b, the four operation modes of TENGs are shown.

There are several examples of TENGs made from lignocellulosic materials. Table 1 shows a list of LC-TENGs classified according to the biomass used, namely cellulose, lignin, or other types of lignocellulosic material. Another group is formed by the LN-TENGs made from non-refined (or poorly refined) lignocellulosic biomass. In addition, the properties of TENGs based on other biomaterials, such as gelatin, alginate, polylactic acid and natural rubber, are also listed for comparison. The voltage, current, and power generated by the LC-TENGs are utilized to document their electrical output performance. As it is usual for TENGs, the generated voltage can achieve high values (700 V in the case of a Pistachio-based TENG), but it is accompanied by low current values (pico and milliamperes). As energy generation is dependent on the surface area of the active components, the specific power density, which is expressed as power per unit area, can be utilized to compare the performance of various TENGs. Table 1 shows that the power density of LC-TENGs ranges from 0.19 mW/m^2^ to 35,100 mW/m^2^. The highest power density was achieved by TENGs made from half-cell allium plant skins. Other high power densities are reported for TENGs made from a cellulose fibers (CF)/natural rubber (NR) composite (3650 mW/m^2^) and black walnut (3800 mW/m^2^). These high power density values are similar to other values reported for TENGs made from other biomaterials and synthetic polymers such as Polyvinylidene Fluoride (PVDF) and poly(trifluoroethylene) (PTrFE) [70]. It should be noted that the electrical performance of TENGs depends on several factors, not just the material used to prepare the active surfaces. It also depends on the roughness of the surfaces, the working mechanism, and the surface chemical modifications, among other factors.

In the case of biomaterial-based TENGs, not only does the electric performance have to be assessed, but so does their environmental impact. Life cycle assessment (LCA) is one of the most useful tools used to assess the environmental impacts related to the entire production chain of a product. LCA allows for the evaluation of the product’s life-cycle environmental impact from the raw material to the final disposal. A great number of papers report on the LCA of various energy harvesting technologies. However, there are few studies on the LCA of TENGs. Ahmed et al. [102] reported on an LCA of TENGs with active surfaces made from acrylic and polytetrafluoroethylene sheets. They found that the requirement for producing these synthetic polymeric sheets is the major contributor to the majority of CO_2_ emissions for TENGs. They also found that these synthetic polymer-based TENGs generally have a better environmental profile than commercialized Si-based and organic solar cells. The utilization of non-refined cellulosic materials could eliminate laborious and energy-intensive processes, thereby minimizing the environmental footprint of LC-TENGs. Another strategy could be the use of recycled materials, extending the life cycles of lignocellulosic products, and, thus, reducing the environmental impact of their use. Zhang et al. [103] utilized wastepaper from the printing industry to create a groundbreaking LC-TENG which achieved a remarkable maximum output power density of 25,500 W/m^2^. LCA studies of LC-TENGs are still needed in order to precisely evaluate their environmental impact.

### 4.1. Refined Lignocellulose-Based TENGs

#### 4.1.1. Cellulose Based-TENGs

Yao et al. [71] fabricated TENGs using cellulose nanofibrils (CNFs). CNF hydrogel was first prepared from bleached kraft eucalyptus pulp. The wood pulp was oxidized and mechanically homogenized following Saito’s method to achieve a dispersion of CNFs. Flat, transparent, and flexible CNF films were obtained after the dilution, filtration, and drying processes under pressure from the CNF hydrogel. The thickness of the CNF film was controlled by varying the amount of CNF hydrogel. The surface roughness of this film was measured to be approximately 300 nm. CNF film and fluorinated ethylene propylene (FEP) were used as positive and negative triboelectric materials. These films were cut into square shapes and attached to the center of an ITO and PET substrate, which served as the electrode and support, respectively. The TENGs exhibited an open-circuit voltage that was dependent on the active surface area. The highest electrical output was achieved with a superficial area of 40 cm^2^, resulting in a short-circuit current (Isc) of 35 µA and an open-circuit voltage (Voc) of 30 V. This was higher compared to the lower area TENGs, such as 16 cm^2^ (Voc =15 V) and 9 cm^2^ (Voc = 7.5 V). The maximum electrical power of 0.56 mW was achieved with a surface area of 40 cm^2^, a thickness of 70 μm, and an external load of 1 MΩ. Figure 6a demonstrates the utilization of this TENG for capturing human step motion.

Saqib et al. [72] designed a novel particle bio-TENG (pTENG) that can harvest energy from all directions of movement. This TENG (Figure 6b) uses commercially available tiny cellulose particles (approximately 6µm in diameter) as the triboelectric positive material. These particles are contained within a gelatin capsule, which serves as the triboelectric negative material. The weight of this pTENG was approximately 0.121 g. Therefore, due to the small size and low weight of the pTENG, it is convenient to aggregate it in 16 units or more. The maximum voltage generated increased from 15 to 85 V, the short-circuit current increased from 409 to 1326 nA, and the maximum power increased from 5.488 to 70 μW when the TENG system was scaled up from 1 to 16 units.

Roy et al. [73] improved the triboelectric properties of cellulose nanofibers (CNFs) prepared using hardwood pulp (kraft pulp) by adding allicin, which is extracted from garlic juice. Allicin (diallylthiosulfinate) is an organosulfur compound found in garlic. The compound responsible for the aroma of garlic is formed via the enzymatic transformation of allicin by the enzyme alliinase. It was grafted onto the surface of the CNFs using a thiolene “click” chemistry. The surface of the CNFs was pretreated with a silanization process using 3-Mercaptopropyl trimethoxysilane (MPTMS) to attach thiol groups. The click chemistry was possible because of the double bonds in allicin and thiol groups. For this purpose, 1.0, 2.5, and 5.0 mL of garlic juice were used to coat the surfaces of pretreated CNF to create films named Alc-S1-CNF, Alc-S2.5-CNF, and Alc-S5-CNF, respectively. Some 2 × 2 cm^2^ TENGs were fabricated using the Alc-Sx-CNF film and PVDF film as triboelectric positive and negative materials. They were attached to aluminum-coated PET and spaced 2 mm apart. The best TENG performance was achieved by the Alc-S5-CNF-based TENG when a force of 12 N was applied and a frequency of 20 Hz was used. This TENG exhibited a Voc of 7.9 V and an Isc of 5.13 µA, which is higher than that of the pristine cellulose-based TENG (1.23 V, 0.80 µA). The current density and power density also increased from 0.2 to 1.28 µA/cm^2^ and 0.25 to 10.13 µW/cm^2^, respectively, in comparison to the pristine cellulose-based TENG. These improvements were explained due to the higher surface polarity, electron-donating capacity, and surface roughness of the Alc-S5-CNF film. These characteristics are attributed to the presence of high dipolar sulfoxide groups (-S=O) with a negative charge centered on oxygen. This TENG was tested under different conditions, demonstrating excellent stability against humidity (RH 25%, 40%, and 60%), long-term cyclic load (over 7000 cycles without a significant drop), and environmental aging (over 12 h of continuous operation).

Zhang et al. [74] inserted silica nanoparticles (Si NPs) into the surface of cellulose nanofibrils (CNFs) to develop an enhanced triboelectric film. They first purified bagasse pulp by bleaching and then ground the cellulose with an ultrafine grinder to obtain cellulose nanofibers (CNFs). Methyltrimethoxysilane (MTMS) and hexadecyltrimethoxysilane (HDTMS) were utilized to cross-link the CNFs with Si NPs, thereby enhancing the hydrophobicity and micro/nano structure of the CNF/Si NP composite film. This composite film exhibited increased hydrophobicity, as evidenced by a water contact angle of 154.7°. It also displayed a surface roughness with an RMS value of 72.61 and low surface energy. The hydrophobic property of the modified CNF-based film can be explained by the presence of silanol groups, which contain both hydrophilic groups (-OH) and hydrophobic groups (-CH3). CNF/TMS/Si NP-based film and fluorinated ethylene propylene (FEP) film were used as triboelectric positive and negative materials, respectively. This TENG unit, with a gear-like structure, was based on a vertical contact separation operating mode. The TENG unit with one gear exhibited a Voc of ~70 V and Isc of ~3 µA. A TENG unit with five gears exhibited a Voc of ~120 V and an I_SC_ of ~3 µA. The spacing between the layers also had a crucial effect on the electrical output. The best performance was achieved with a 3 mm distance, resulting in a Voc of ~125 V and an Isc of ~7 µA. The RT-TENG integrates three TENG units and exhibited an output voltage of 125 V and an Isc of 6.1 µA. It also had cycling stability over 10,000 cycles.

Lin et al. [75] mixed polyethylene oxide (PEO) with cationic cellulose fibers to prepare PEO/cellulose composite paper (PEO/CCP). Commercially available wood dissolving pulp was used as a source of cellulose. Cellulose fibers were first cationized via a reaction with aminoguanidine hydrochloride. Then, PEO was dissolved in a cationic cellulose suspension and freeze-dried to obtain a PEO/cellulose aerogel. This aerogel was then compressed to prepare the PEO/CPP samples. PEO/CCP was used to fabricate a TENG with higher electrical output compared to a TENG based on cellulose paper. PEO/CCP was obtained after compressing a composite aerogel of PEO and cationic cellulose fibers (PEO/CCF aerogel). This aerogel was obtained after a subsequent process, which included dialdehyde modification, aminoguanidine hydrochloride (AGH) treatment, and PEO compounding modification. The weight ratio of Sodium periodate (NaIO4) to cellulose fibers was set at 1:5, 1:3, 1:2, 1:1, and 2:1 in order to regulate the reaction between the aldehyde group and aminoguanidine hydrochloride and thus control the content of amino groups. Cationic cellulose fibers (CCFs) obtained from the reaction were labeled as CCF-1, CCF-2, CCF-3, CCF-4, and CCF-5, respectively. PEO was added to the suspension of cellulose fibers, and later the mixture was freeze-dried in a lyophilizer to obtain PEO/CCF aerogel. PEO/CCP-x was obtained after compressing PEO/CCF-x. Some 3 × 3 cm^2^ TENGs were fabricated using cellulose composite paper-based film as the triboelectric positive material. Some negatively charged triboelectric materials, such as PDMS, PTFE, and FEP, were tested. Ag and Al were used as the electrodes for PEO/CCP and PDMS, respectively. The PEO/CCP-4 with PDMS exhibited the best performance at a frequency of 3 Hz and an external force of 40 N, with a Voc of 222.1 V, an Isc of 4.3 µA, and a Jsc of 39.4 μC/m^2^. The triboelectric performance of the PEO/CCP-based TENG had a linear dependence on the amino group content of PEO/CCP. The performance of the PEO/CCP-4 TENG, when paired with PDMS, was higher than that of the pure PEO TENG or pure CCP TENG (141.1 V and 1.7 µA). It also outperformed the pure cellulose paper (CP) TENG (100 V and 1 µA), as well as the PEO/CCP-4 TENG paired with PTFE or FEP (100 V and 4 µA). The improvement of TENG performance via the use of PEO/CCP-4 can be attributed to the presence of amine groups and the introduction of PEO, both of which enhance the electron-donating property of cellulose. This TENG was also tested under high concentrations (ranging from 50% to 90%) of relative humidity (RH), demonstrating a higher output even at 90% RH with a Voc of 131.5 V and an Isc of 2.6 µA, compared to a pure cellulose-based TENG (~20 V and ~0.3 µA), at the same conditions. The PEO/CCP-4 TENG achieved a maximum power density of 217.3 mW/m^2^ at a loading resistance of 60 MΩ. This TENG also demonstrated long-term stability, with no noticeable decline even after 20,000 contact separation cycles.

Thakur et al. [76] fabricated a three-dimensional (3D) TENG inspired by origami (seen in Figure 6c). They used carboxymethylated cellulose nanofiber (CM-CNF) and perfluoroalkoxy (PFA) films as the triboelectric positive and negative materials, respectively. In order to increase the contact area, two-dimensional (2D) planar structure films were simply folded to obtain a helical 3D structure inspired by origami techniques. Kraft wood pulp was used as the cellulose source to produce cellulose nanofibers (CNFs). Chloroacetic acid was used in the carboxymethylation process, which involves substituting the hydroxyl groups in cellulose with carboxymethyl groups. The carboxymethyl content and the nanofibrillation of cellulose were controlled by reaction time (30 min, 120 min, and 210 min). The diameter of CM-CNFs decreased from 10 nm to 5 nm as the carboxymethylation reaction time increased from 0 to 210 min. Not only was the carboxymethyl group inserted, but Na+ ions were also incorporated into cellulose nanofibrils. The carboxylate content increased from 0 µmol/g for pristine CNFs to 750 µmol/g after 210 min of reaction time. The surface charges also increased from -30.98 mV for pristine CNFs to -73.6 mV after 210 min of reaction time. The morphology of CNF was also enhanced after film casting, transitioning from a rough film with small cracks to a smooth film without any cracks. These enhancements in surface charges and morphology played a vital role in increasing the surface charge density. In order to construct the 3D multi-layered TENG, strips of CM-CNF film and PFA film measuring 20 × 2.5 cm^2^ were cut and then attached to both sides of a copper electrode using carbon tape. This TENG was operated in contact–separation mode, and its total contact area in the 3D structure was around 100 cm^2^. The CM-CNF210/PFA TENG achieved the best TENG performance with a Voc of 125 V and an Isc of ~12µA under 10 N of load and 2 Hz of frequency. These values were higher than those of the pristine CNF TENG (38 V and 4 µA), CM-CNF120/PFA TENG (80 V and 8 µA), and CM-CNF30/PFA TENG (70 V and 7 µA). The CM-CNF210/PFA TENG had a maximum power density of 2076 µW with an external load of 10 MΩ. This TENG also demonstrated good stability, with no loss in output performance over 10,000 cycles. The effect of the increase in frequency and external load was also studied, showing a direct relationship with the electrical outputs of the TENG.

Chenkhunthod et al. [77] inserted cellulose fibers (CF) and Ag nanoparticles (Ag NPs) into natural rubber (NR) to enhance its electron-donating ability. Cellulose microfibers were synthesized from sugarcane leaves (SL). Various molar concentrations (1, 2, and 3 mM) of AgNO_3_ were used to produce different quantities of nanoparticles on the surface of CFs. These Ag–CF hybrid materials were named CF@Ag_1, CF@Ag_2, and CF@Ag_3, respectively, based on the molar concentrations of AgNO_3_. These fillers were mixed with natural rubber latex (0.20 wt%) to obtain NR-Ag@CF films via the casting and drying process. The silver (Ag) contents in the CF@Ag_1, CF@Ag_2, and CF@Ag_3 were 1.37%, 2.22%, and 4.38% by weight, respectively. PTFE film and NR composite film were used as the negative and positive triboelectric materials, respectively. Some single-electrode mode TENGs with a contact area of 4 × 4 cm^2^ were fabricated using these films and an ITO film as the electrode. The best output performance was achieved with the NR-CF@Ag_3-based TENG, which exhibited an electrical output voltage (Vpp) of 128 V and an electrical output current (Ipp) of 12.4 µA under an external force of 5 N at a frequency of 5 Hz. These values were higher than those of the pure NR-based TENG (58 V and 5.8 µA), NR-CF-based TENG (104 V and 9.8 µA), NR-CF@Ag_1 (112 V and 10.2 µA), and NR-CF@Ag_2 (120 V and 11.8 µA). The dielectric constants of these films were measured, showing an inverse correlation with the number of Ag NPs. The NR-CF@Ag_3 exhibited the lowest dielectric constant (~6.5), while the pure NR film has a dielectric constant of 11. This was explained by the absence of a capping agent during the synthesis of Ag nanoparticles. It allows the connection between the particles, forming an electrical conductive path and reducing the dielectric constant. The increase in electrical output with the addition of Ag NPs can be attributed to the intrinsic properties of Ag nanoparticles, which enhance electron-donating ability in Ag–cellulose-filled NR films. The NR-CF@Ag_3 TENG exhibited a maximum power density of 3.65 W/m^2^, which was higher than that of the NR-CF TENG (2.65 W/m^2^) and the pure NR TENG (0.68 W/m^2^), at a resistance of 0.7 MΩ. The effect of frequency and external load was also studied, revealing a direct relationship with the electrical outputs of the TENG. This TENG also demonstrated an 82% output retention after 10,000 cycles.

Varghese et al. [78] used electrospun cellulose acetate nanofibers and surface-modified PDMS to fabricate a cellulose-based TENG with an improvement in electrical output. Two types of micropatterns (microdome-structured and micropyramid-structured) were developed in PDMS using silicon inverted molds. Cellulose acetate (CA) is a non-toxic and biodegradable material derived from cellulose. The triboelectric material was deposited onto the electrode using the electrospun technique to increase its effective area. For this purpose, CA was electrospun in an acetone/DMF mixture solution at a flow rate of approximately 1 mL/h to produce nanometer-sized fibers. The effective contact area increases due to the synergistic improvement in contact area provided by the microstructure pattern in PDMS and electrospun CA nanofibers. It resulted in the enhancement of triboelectric properties of the TENG devices. The microstructures patterned in PDMS did not damage the nanofibers to the extent that it affected the output performance. The TENG with pyramidal surface modification achieved the best performance (Figure 6d). This TENG displayed a Voc of. ~400 V, a Jsc of 3 mA/m^2^ with an input force of 3 N at a frequency of 4 Hz. These values were greater than those of the TENG with dome surface modification (105 V and 0.9 mA/m^2^) and the plain PDMS TENG (75 V and 0.5 mA/m^2^). This TENG also exhibited the highest maximum power density at 900 mW/m^2^, with a load resistance of 10^8^ Ω. It had a higher power density than the plain TENG and the dome TENG, which reached their maximums at ~5 mW/m^2^ and 7 mW/m^2^, respectively.

#### 4.1.2. Lignin-Based TENGs

Lignin is the second most abundant biopolymer on Earth and one of the least utilized plant components. Lignin is primarily produced by the pulp and paper industry on a scale of approximately 100 million tons per year. Only around 5% of lignin production is currently utilized in applications such as adhesives, binders for animal feed, bricks, and ceramics. The remaining lignin is primarily incinerated as a cost-effective fuel. Lignin is a water-insoluble, biodegradable, and biocompatible component of lignocellulosic material and possesses an irregular 3D structure. Bao et al. [79] fabricated the first lignin-based TENG. For this purpose, kraft lignin was mixed with starch because the insolubility of lignin fibers prevents the formation of a uniform film or gel on its own. The addition of an agent, such as gelatin and sodium hydroxide (NaOH), has a plasticizing effect. Inherently, lignin possesses a strong ability to donate electrons, while Kapton has a strong ability to accept electrons. The surface roughness increased with a higher concentration of lignin. The cross-linking of the starch matrix was disrupted due to the presence of hydrophobic lignin particles. This increase in roughness had a beneficial effect on the surface area, resulting in an increase in the electrical TENG output. The TENG based on a concentration ratio of 3:7 for lignin to starch exhibited the best performance among the TENGs made solely from starch and lignin. This TENG exhibited a Voc of 1.04 V/cm^2^ and an I_SC_ of 3.96 nA/cm^2^. Concentrations beyond this weight ratio tended to decrease the electrical output. An excess of lignin caused aggregation in the form of clusters, which increased conductivity but reduced the charge induced in the electrodes. The inclusion of glycerol significantly enhanced the strength and flexibility of the lignin composite film. However, it also resulted in a polarity reversal of the TENG. The explanation is that the OH groups in glycerol interact, via hydrogen bonds, with H_2_O molecules from the casting solution. The presence of more OH groups, resulting from the addition of glycerol, enhances the film’s electron-attracting capability. This is because these functional groups have a high electronegativity. This strong ability to capture electrons was even higher than that of Kapton. As a result, the electrical output was reversed. The TENG based on a 6% glycerol in a 3:7 lignin/starch composite film exhibited the best performance, with a Voc of −2 V/cm^2^ and an Isc of −13 nA/cm^2^. The addition of NaOH also had a positive effect on the electrical output. The best performance was achieved with the TENG based on a 0.5 M NaOH, 6% glycerol, and 1:9 lignin-starch composite film, among different molar concentrations of NaOH. This TENG exhibited a Voc of −3.5 V/cm^2^, an Isc of −23 nA/cm^2^, and a maximum power density of 173.5 nW/cm^2^. This TENG also demonstrated good electrical stability over 1800 cycles of contact–separation.

An et al. [80] developed TENGs based on nanofibers (NFs) fabricated using a solution blow-molding process. These nanofibers were made from solutions containing soybean protein and lignin. Each solution was then deposited onto a Cu film, which served as the electrode. Soybean protein and lignin-based films were used as triboelectric positive materials, while polyimide film was used as the negative material. Both TENGs demonstrated superior performance compared to other deposition methods, such as electrospinning and brush-casting, when applied to nylon 6. The SB-soy protein and lignin NF-based TENGs exhibited rectified voltages of 0.7 V and 4.5 V, respectively, at a frequency of 10 Hz and a force of 21.5 N.

Sun et al. [81] fabricated a conductive and transparent organohydrogel with fast gelation using a lignin-based self-catalytic system. This conductive organohydrogel was used as an active triboelectric material, which also served as an electrode in TENGs. The presence of abundant methoxyl (–OCH_3_) and phenolic hydroxyl (–OH) groups in lignin enables the formation of a self-catalytic system with metal ions, leading to rapid polymerization. This self-catalytic system is based on the participation of polyphenols from alkali lignin (AL) and the metal ions Cu^2+^. The AL is a water-soluble modified lignin that naturally dissolves in alkaline environments. Ethylene glycol (EG) was used as an effective inhibitor for water freezing, but it also accelerated polymerization. The synergistic effect of both the material and the process promoted the formation of multiple intermolecular hydrogen bonds between EG and water molecules. The resulting organohydrogel exhibited an interfacial adhesion of up to 31.4 kPa, an excellent freezing ability as low as -40 °C, a superior water retention capacity, and a good stretchability (approximately 800%). This organohydrogel, with a water-to-EG weight ratio of 2:3, was utilized in the production of a single-electrode TENG (O-TENG). The organohydrogel, which had a thickness of 1 mm and a surface area of 2 × 3 cm^2^, was used as the working electrode. It was sandwiched between two pieces of VHB film. The VHB/organohydrogel was used as the triboelectric negative material, while a copper wire was directly connected to the AL-Cu@W/EG-PAM hydrogel for electrical connection. Al-foil paper, Cu film, Kapton, PU film, paper, and polyethylene terephthalate (PET) film were used as the counterparts. The electrical output using these films was in the following order: Al > Cu > Kapton > PU > paper > PET. The best performance was achieved with the use of an Al layer as the triboelectric positive material. This TENG (Figure 7a) exhibited a Voc of 220 V and an Isc of 4.5 pA. When the hydrogel was stretched by 100%, the electrical output of the O-TENG increased to 330 V. One possible explanation for this is that stretching the O-TENG enlarges its surface area, thereby increasing the surface charge density. This TENG also exhibited good stability without any output decay over more than 6000 cycles.

Funayama et al. [82] fabricated a metal-free and biodegradable TENG by utilizing laser-induced graphitization on a film made of lignin/poly (L-lactic acid) (PLLA). PLLA is a biodegradable polymer synthesized from biomass materials. Laser-induced graphitization was used to fabricate conductive structures composed of graphitic carbon onto the surface of a lignin/PLA film via laser irradiation. For this purpose, alkali lignin powder was mixed with PLLA pellets to prepare composite sheets with different lignin concentrations (0, 5, 10, 20, 30, 40, and 50 wt%). Many 13 × 13 mm^2^ films were fabricated to study the effect of lignin concentration, laser power, and laser scanning speed on electrical conductance. Setting the laser power and scanning speed at 200 mW and 200 μm/s, we studied the effect of lignin concentration. A black structure was observed in films containing more than 10 wt% lignin, confirming the formation of carbonized structures. For lignin concentrations exceeding 40 wt%, holes with a diameter of 20 μm were observed on the surface of the film. The best conductance of ~2.4 mS was obtained from a concentration of 30 wt% lignin. The effect of laser power was studied by setting the laser scanning speed at 200 μm/s and the concentration of lignin at 30 wt%. The best conductance, ~2.5 mS, was achieved with the use of 200 mW laser power. The effect of laser scanning speed was studied by setting the laser power at 200 mW and the lignin concentration at 30 wt%. The best conductance of ~2.5 was achieved at a laser scanning speed of 200 μm/s. A TENG was fabricated using a lignin/PLLA composite film. The film was made with 30 wt% lignin, 200 mW of laser power, and a speed of 200 μm/s. A PDMS sheet was used as the triboelectric positive material, while another PDMS sheet served as the triboelectric negative material. This TENG (Figure 7b) exhibited a Vpp of ~14, 15, and 20 V under 1, 10, and 100 N of force applied, respectively, at a frequency of 1 Hz. This TENG exhibited a maximum power density of ~1.98 mW/m^2^ with a load resistance of 200 MΩ when a force of 1 N was applied.

#### 4.1.3. TENGs Based on Other Lignocellulosic Components

Alluri et al. [83] used aloe vera Barbandensis Miller (AV) as a triboelectric material in the fabrication of TENGs. AV film was fabricated by depositing and spin-coating a liquid AV gel (AV parenchyma) onto the surface of the aluminum (Al) electrode. AV material contains both positive and negative charges due to the presence of acemannan, pectin, and other components such as carbohydrates, proteins, and minerals (primarily cations). Some TENGs with a contact–separation mode of 3 × 3 cm^2^ were fabricated using the AV film as the triboelectric positive material. Paper, PVC, and PDMS were tested as potential triboelectric negative materials and were attached to a Au electrode. The best performance was achieved by the AV film/PDMS-based TENG (Figure 8a) with a Voc of 32 V and Isc of 0.11 µA at an acceleration of 2 m/s^2^ of the linear motor. These values were higher than those of the AV/PVC-based TENG (15 V and 40 nA) and AV/paper-based TENG (10 V and 40 nA). The AV film/PDMS-based TENG had a maximum instantaneous power density of 0.19 µW/cm^2^ at a load resistance of 100 MΩ. This TENG also demonstrated good stability, with no significant change in output voltage observed over a period of 1300 s. Alluri et al. [83] also fabricated a free-standing mode TENG composed of AV liquid gel as the triboelectric positive liquid material. FEP and PDMS were tested as potential triboelectric solid negative materials. Gold (Au) was sputter-coated onto the triboelectric negative material, and the composite was glued to the internal wall of a tube. The AV was diluted in polar ethanol (ETH) to reduce the viscosity, and it was later introduced into the tube. This TENG could harvest the mechanical movement of the AV gel caused by manual force excitation. The device area was approximately 2.7 cm^2^, and the AV gel-PDMS TENG (Figure 8a) achieved the best performance. This TENG exhibited a Voc of 12.72 V and an Isc of 113.23 nA, higher than the other AV gel-FEP TENG (0.6 V and 20 nA).

Zhang et al. [84] utilized the dual affinities of half-cell allium plant skins to construct single-material TENGs (SM-TENGs). Allium plants, such as leek, scallion, and onion, were used as both positive and negative triboelectric materials. To evaluate the chemical composition and triboelectric properties of the inner and outer surfaces, the monolayer skin cells of plants were cut, and their composition was verified using SEM. FTIR characterization showed that the outer surfaces were rich in C–H, C=O, C–C, and -OH groups in all cases. However, leek skin also exhibited –NH_2_ or –NH groups on the surface. SM-TENGs were fabricated using the inner and outer surfaces of each plant. A leek-based TENG achieved the best performance, with a Voc of 182 V, an Isc of 0.83 mA/m^2^, and a power density of 35.1 W/m^2^. These electrical outputs were higher than those of the scallion half-cell skin-based TENG (60 V, 0.25 mA/m^2^, and 4.1 W/m^2^), the onion half-cell skin-based TENG (60 V, 0.25 mA/m^2^, and 4.1 W/m^2^), and other TENG combinations.

Babu et al. [85] extracted leaf powder from the Rumex vesicarius plant to fabricate a vertical contact–separation mode TENG. Leaves were washed and dried before being macerated using non-polar and polar solvents to obtain a powdered extract. A solution mixture of ethanol and deionized water in a 1:1 ratio was used to disperse the powdered leaves. This final solution was drop-cast and dried onto aluminum foil, which served as the electrode in the TENG. A TENG was fabricated using a film made from this leaf powder and PET as the triboelectric positive and negative materials, respectively. PET was attached to ITO, which served as the other electrode. The 5 × 5 cm^2^ leaf powder/PET-based TENG device (Figure 8b) exhibited an open-circuit voltage (Voc) of 3.86 V and a short-circuit current (Isc) of 3.78 µA under biomechanical (hand slapping) force. This TENG had a maximum power density of 0.1894 µW/cm^2^ at load resistances of 20 MΩ, and it demonstrated good stability over 1200 cycles.

Feng et al. [86] utilized Syringa vulgaris leaves and leaf powder to construct 4 × 4 cm^2^ TENGs in vertical contact–separation mode. Fresh leaves, dried leaves, and leaf powder were used as triboelectric positive materials. PVDF was used as the negative triboelectric layer. Both layers were attached to a Cu film, which served as the electrode. The fresh and dried leaf-based TENGs exhibited a Voc of 430 V and 560 V, respectively, and an Isc of 15 µA and 25 µA, respectively, at a contact frequency of 5 Hz. This lower performance in fresh leaves was explained by the reduction in electron generation caused by the presence of water. The leaf powder-based TENG (LP-TENG) exhibited higher performance with a Voc of 660 V and an Isc of 26 µA. The increase in the contact area is explained by the micro particle surface structure. Poly-L-Lysine (PLL) was used to improve the output performance of a film made from leaf powder. The leaf powder-PLL (LP-PLL) TENG (Figure 8c) exhibited a Voc of 1000 V and an Isc of 60 µA at a frequency of 5 Hz. This TENG had a maximum power of 17.9 mW with a loading resistance of 11 MΩ. This increase in performance was explained by the fact that PLL changed the surface composition, thereby enhancing the electrical output.

### 4.2. TENGs Based on Non-Refined Lignocellulose

The utilization of lignocellulosic biomass in TENGs, without the requirement for extensive chemical/mechanical refining, could be crucial in achieving a reduced environmental impact and lower-cost production compared to other lignocellulosic-based TENGs that rely on harmful chemicals or laborious and costly processes. Shi et al. [104] fabricated a low-cost, biodegradable, and recyclable TENG using lignocellulose bioplastic (LB). LB was produced from powdered lignocellulosic biomass. The biomass used was obtained from poplar wood, wheat straw, rice straw, corn straw, bagasse, and bamboo. The LB fabricated from these biomass sources showed similar results. LB was manufactured using an in situ lignin regeneration and cross-linking modification approach. A deep eutectic solvent (DES) composed of choline chloride (ChCl) and oxalic acid was utilized to dissolve the lignin and break down the wood by disrupting the hydrogen bonding among cellulose fibers. Lignin was regenerated in situ by adding water. Lignin was bound to the interconnected cellulose micro/nano fibril network via the formation of hydrogen bonding. Citric acid (CA) was then used as a bond promoter to enhance the cross-linking of lignin and cellulose. CA contains a large number of carboxyl and hydroxyl groups, which not only cause the self-cross-linking of lignin and cellulose, but also facilitate cross-linking via the formation of ester bonds between lignin and cellulose. This method eliminates the need to separate and isolate lignin and cellulose. A conductive LB film (C-LB) was produced by incorporating carbon powder into the LB manufacturing process. The LB film demonstrated an enhanced tensile strength of 99 MPa, in comparison to that of the cellulose film at 10 MPa. LB can be completely degraded on the soil surface within 12 days and inside at a depth of 5 cm within 2 days. LB stored in a real work environment did not show any changes even after more than 10 months. At end-of-life, LB can be recycled and used again in the production of AL-TENG via a simple mechanical cutting and stirring process. The single-electrode mode AL-TENG with an area of 2 × 2 cm^2^ was developed using LB and PTFE as the triboelectric positive and negative layers, respectively. The C-LB film was used as an electrode, and the contact pressure used was 3.5 kPa. This TENG exhibited a Voc of 31 V, an Isc of 0.2 µA, and a transferred charge (Qsc) of 8 nC. The maximum output power density of 10 mW/m^2^ was achieved with an external load resistance of 80 MΩ. The increase in frequency and the number of contact–separation cycles did not have a significant effect on the electrical output of this TENG. It was recycled and reprocessed up to five times with a negligible change in TENG performance. The AL-TENG was proposed for use in the field of intelligent disposable medical monitoring.

Yao et al. [71] recycled biodegradable paper cardboard and used it to create a power board based on TENGs. First, the cardboard was dispersed and agitated in water. Into this wet mixture, an 8 × 8 cm^2^ CNF-based TENG was embedded. This composite was vacuum-sealed, hardened via cold pressing at 100 MPa for 1 h, and dried at 65 °C under 50 lb of pressure for 24 h to produce a fiberboard with a diameter of 20 cm and a thickness of 4 mm. This TENG-integrated fiberboard was placed on the floor to harvest mechanical energy from human steps. A single human step generates an open-circuit voltage between 10 and 30 V, and a short-circuit current between 10 and 90 µA. It easily turned on 35 green LEDs. This electrical output represented up to 98% of the triboelectric charge to the external circuit, with a conversion efficiency of 8.3% from input mechanical energy. This low conversion efficiency can be explained by the fact that mechanical energy is mostly stored as mechanical deformation due to the rigidity of the fiberboard.

Zhou et al. [87] used various types of treated wood to develop TENGs in vertical contact–separation mode. Natural veneers were extracted from various types of trees, including Fraxinus mandshurica, black walnut, cherry, and golden teak (referred to as NW1, NW2, NW3, and NW4). These veneers had a thickness of 500 µm and were used to create positive triboelectric films based on wood. These veneers were first pretreated to develop a dense structure via chemical treatment and hot pressing, which increased the contact area. This chemical treatment removes non-structural components, such as lignin, hemicellulose, and wood extracts, by immersing the material in a boiling NaOH/Na_2_SO_3_ solution. This pretreatment process also improved the mechanical strength of the wood, increasing the Young’s modulus from 7 GPa (for untreated veneers) to 11 GPa. It improved flexibility and hydrophobicity by promoting the orientation of the cellulose structure. These films were named HPW1-4 for the veneers NW1-4, respectively. The post-modification method used to pretreat wood veneer-based films was inspired by the adhesion of dyes to the surface of fabric. This surface treatment involved cationic modification using 3-chloro-2-hydroxypropyl trimethylammonium chloride (CHPTAC) via the solution immersion method. This process improved the surface potential from 8.6 mV, which was related to the NW, to 18.66 mV via the adhesion of cationic groups, resulting in the formation of a positive triboelectric layer. These films were named MPW1-4 for NW1-4, respectively. Some TENGs were fabricated using wood-based films and polytetrafluoroethylene (PTFE) film as the positive and negative triboelectric layers. These TENGs were then attached to acrylic plates using nickel tape. TENGs based on HPW1-4 exhibited Isc of 6.54, 4.54, 5.27, and 6.58 μA, respectively. TENGs based on MPW1-4 exhibited Isc of 9.33 μA, 9.74 μA, 7.04 μA, and 8.83 μA, respectively. The best performance of this modified pressed wood-based TENG (Figure 9a) was achieved via the synergistic effect of both treatments, resulting in a high open-circuit voltage, short-circuit current, and peak power density of 335 V, 9.74 μA, and 3.8 W/m^2^, respectively.

Zhang et al. [88] utilized the lignocellulosic structure of Nanzhu bamboo as a template for depositing Ti_3_C_2_Tx nanosheets, resulting in the development of highly efficient and moisture-sensitive triboelectric materials. Lignin and hemicellulose were partially removed from natural bamboo during the delignification process to create a multilayer porous cellulose template. The monomer Ti_3_C_2_Tx was selected as a filler due to its high specific area, which is attributed to the presence of hydrophilic functional groups -OH and -O, as well as its high electrical conductivity. Due to the abundant presence of functional hydroxyl groups (-OH), cellulose offers numerous active sites for Ti_3_C_2_Tx attachment. In the depressurized impregnation process, cellulose was impregnated with the monomer Ti_3_C_2_Tx via the formation of hydrogen bonds. The surface of cellulose bamboo was severely damaged due to prolonged delignification, which has an impact on its mechanical stability. Electrical conductivity, pore size, and porosity of cellulose/Ti_3_C_2_Tx composite films were evaluated. The best performance was achieved with 8 h of delignification time. It reached a maximum conductivity of 2.07 S/cm, a pore size of 613.43 nm, 71.83% pore density, and a pore volume of 0.40 g/mL. The composite film with 10 h of delignification showed lower conductivity than the others with a shorter processing time. With a prolonged delignification time, the cellulose began to undergo hydrolysis. Cellulose/Ti_3_C_2_Tx composite and fluorinated ethylene propylene (FEP) membranes were used as positive and negative triboelectric materials in the construction of TENGs. The cellulose/Ti_3_C_2_Tx composite film (Figure 9b) achieved the best output performance, with a delignification time of 8 h and a thickness of 1 mm. This TENG exhibited a Voc of 60 V, an Isc of 10 µA, and a transferred charge of 28 nC. The maximum power density was 25 µW/cm^2^ with an external resistance of 7 × 10^5^ Ω.

Dudem et al. [89] fabricated a TENG using waste and recyclable materials, such as paper and plastic. Paper wipes (PWs) recycled from the waste bin were coated with carbon nanoparticles (CNPs) to enhance conductivity and decrease sheet resistance. CNPs were coated onto micro-fibrous-networked PWs using a simple brush-painting method, followed by a curing treatment. This C@PW film served as both an electrode and a positive triboelectric material. Different concentrations of carbon nanoparticles (NPs), ranging from 10 to 16 wt%, were evaluated to measure sheet resistance and TENG performance. The sheet resistance decreased with the increase of carbon nanoparticles. FE-SEM images showed an adhesion of 60 nm carbon nanoparticles to each microfiber. Waste plastic coffee cups made from polytetrafluoroethylene (PTFE) were collected to serve as a negative triboelectric material. Some TENGs with an area of 25 cm^2^ were fabricated using both carbon@polytetrafluoroethylene (C@PTFE) and polytetrafluoroethylene (PTFE) film as triboelectric materials. C@PW also served as an electrode for TENGs. The short-circuit charge (Qsc), Isc, and Voc corresponding to C@PW-TENGs were gradually increased with the concentration of CNPs reaching the maximum values of 1.56 µA, 78 nC, and 80 V with 15.13 wt% of CNPs (Figure 9c) under the conditions of 25 N and 1 Hz of force and frequency, respectively. The increase in TENG output values was explained by the increase in effective surface area resulting from the insertion of carbon nanoparticles. TENGs with CNPs beyond this concentration exhibited lower performance. This can be explained by surface contamination or agglomeration in the surface by NPs. The TENG performance also showed a linear dependence on the applied force and frequency. The maximum power achieved was 0.58 mW, with a resistance of 100 MΩ, a force of 30 N, and a frequency of 1 Hz. The durability was tested, showing a relatively stable output after 18 days with 10,000 cycles per day.

Saqib et al. [90] used lignocellulosic waste fruit shells (WFSs) in the development of TENGs. Lignocellulose waste from almond (A), walnut (W), and pistachio (Pi) shells were used as positive triboelectric materials. Each type of WFS possesses different concentrations of lignin and cellulose, which are the main compounds. These concentrations also caused different levels of oxygen due to the varying number of hydroxyl groups. A-WFS, W-WFS, and Pi-WS contain 38.47%, 36.38%, and 43.08% cellulose, and 29.54%, 43.70%, and 16.33% lignin, respectively. The cellulose concentrations of A-WFS, W-WFS, and Pi-WS were 23.3%, 25.4%, and 33.7%, respectively. WFSs were purchased in the local market. Later, they were washed with deionized (DI) water and dried at room temperature for 2–3 days. These dried materials were ground three times using a vacuum grinder, and the resulting powder was dispersed onto a glue-coated aluminum conductive electrode. In the presence of oxygen, WFSs have the inherent capability to lose electrons, allowing them to be used as a positive triboelectric material. WFS-based films with dimensions of 4.5 × 4.5 cm^2^ and a thickness of 30 μm were used in the fabrication of TENGs. Each TENG has different counterparts for each WFS film, such as pure cellulose, pure lignin films, another WFS film, and negative triboelectric synthetic materials such as polydimethylsiloxane (PDMS), polyvinylidene difluoride (PVDF), polytetrafluoroethylene (PTFE), and poly(ethylene terephthalate) (PET). Among the comparison of organic and degradable layers, the TENG based on Pi-WFS and W-WFS exhibited the best electrical output performance. It had a Voc of 70 V and an Isc of 3.7 µA, which was higher than the others. The Voc and Isc of the other TENGs ranged from 23 to 76 V and 1.1 to 2.02 µA, respectively. WFS-based TENGs, utilizing synthetic negative triboelectric materials, demonstrated superior performance. The best TENG performance was achieved by using Pi-WFS and PTFE films as positive and negative materials. This TENG exhibited a Voc of 700 V, an Isc of 95 mA, and a maximum power density of 416.14 µW/cm^2^ at a resistance of 3 MΩ, with a force of 75 N applied and a frequency of 10 Hz. Saqib et al. [90] also studied the effect of relative humidity (RH) in the range of 30 to 90% on the WFS-based TENGs. The W-WFS/PTFE-based TENG (Figure 9d) exhibited the best performance under humid environments (30–90% RH). The power of this TENG slowly decreased until it reached 50% RH, and then decreased linearly at higher humidity. The other TENGs showed a linear decrease from 30 to 90% RH. This can be explained by the majority content of lignin being hydrophobic. WFS films with a high cellulose content allow for easy diffusion and absorption of water, creating a barrier that hinders the transfer of electrons.

Zhang et al. [91] used peanut and chestnut shells from agricultural waste to construct a stator/rotator system-based TENGs. Agricultural waste such as nutshells is rich in lignin, cellulose, and hemicellulose. Because of the abundance of hydroxyl groups in these compounds, the shell powder is capable of improving the surface contact area and, consequently, the superficial charge density in triboelectric materials. Nutshell powder was deposited onto an aluminum (Al) film that was attached to a Kapton substrate. The rotor of this TENG was placed in the center of the system; it was composed of PTFE, Al, and PET film. This rotor turned, powered by a motor, at a frequency of 2 Hz. The stators were placed around the rotor and were composed of peanut shell powder (PSP)-based flexible films. PSP and PTFE film were used as triboelectric positive and negative material, respectively. The PSP-TENG system exhibited an Isc of 24.8 μA and a Voc of 171.3 V, higher than the same system without the PSP film (Isc = 10 μA and Voc = 50 V). The PSP/PTFE TENG showed higher performance (Isc = 17.5 μA and Voc = 145 V) compared to other tribonegative layers such as FEP (Isc = 12.5 μA and Voc = 90 V), PVC (Isc = 12.5 μA and Voc = 90 V), PET (Isc = 4.8 μA and Voc = 38 V), and Kapton (Isc = 2.4 μA and Voc = 18 V). The particle size of nutshell powder also affects its triboelectric performance. The output of PSP-TENG increased when the particle size decreased. This can be explained by the fact that small particles result in higher surface roughness and a larger effective friction surface area. The TENG with 250 mesh PSP film exhibited an Isc of 24.8 μA and Voc of 171.3 V while the TENG with 1000 mesh PSP film (Figure 9e) exhibited an Isc of 35 μA and Voc of 225 V. TENG output performance also increased with the frequency used. The best performance was achieved at 5 Hz with an Isc of 42 μA and Voc of 210 V. The 250 mesh PSP TENG at 2 Hz with 8 MΩ achieved the maximum output power density of 365 mW/m^2^.

Bang et al. [92] used balsa wood pieces to produce flexible porous friction layers as both negative and positive materials. Wood was sulfonated using a mixed solution of NaOH and Na_2_SO_3_. Then it was softened by removing partial lignin/hemicellulose via bleaching with H_2_O_2_ solution. The white wood aerogel, C-Wood, obtained had a weight loss of 84% and a dimensional reduction of 39% compared to wood due to the removal of lignin and cellulose. N- (2-aminoethyl) -3-aminopropyltrimethoxysilane (AEAP-Si) and trichloro (1H,1H,2H,2H-perfluorooctyl) silane (PFOT-Si) were used to the chemical graft NH2 and CF3/CF2 functional groups in C-Wood surface via silane reaction. These products, after being pressed to increase density, were used as positive and negative triboelectric material, respectively; they are called N-Wood and F-Wood respectively. Surface modification increased the hydrophobicity of cellulose-based material. It was validated by the measurement of water contact angle. The angle after 5 s of a droplet deposition was 0°, 0°, 116.5°, 89.6° with pristine balsa wood, C-Wood, F-Wood, and N-Wood, respectively. This was due to the high content of fluorine in F-Wood and reduced hydroxyl groups in N-Wood. SEM images and X-ray spectroscopy confirmed the homogeneous distribution of F in F-Wood and N in N-Wood. These films were cut into 2 × 2 cm^2^ pieces to construct various combinations of TENGs: N-Wood//F-Wood, NWood//C-Wood, F-Wood//C-Wood, and C-Wood//C-Wood. All of them were tested at 3 Hz of frequency under 8.2 N of external force and 6 mm of spacer distance. The Voc and Isc values of these TENGs were as follows: C-Wood//C-Wood (4 V, 30 nA), N-Wood//C-Wood (24 V, 99.33 nA), N-Wood//F-Wood (62.5 V, 0.241 μA), and F-Wood//C-Wood (90.1 V, 0.458 μA). The best performance was achieved with a TENG based on F-Wood//C-Wood. This was because the amine group of N-Wood is less electronegative than the hydroxyl group of C-Wood. This TENG was capable of generating 54.5 μW at a load resistance of 47 MΩ.

Tanguy et al. [93] used red cedar bark to produce natural lignocellulosic nanofibrils (LCNFs) derived from wood. A negative triboelectric LCNF film was developed from LCNFs because the presence of lignin on the surface of LCNF imparts strong tribonegative properties due to the presence of C-C bonds. Some TENGs were fabricated with LCNFs as tribonegative material and aluminum films as electrodes. Many triboelectric materials, such as PET, Kapton, ITO, Cu, and Al, were tested under an external force of 20.8 N at a frequency of 5 Hz. LCNF@Cu TENG achieved the best performance with a Voc of 300 V, an Isc of 11.1 µA/cm^2^, and an output power density of 52 µW/cm^2^.

Sun et al. [68] modified the surfaces of three wood species (balsa, spruce, and yew) to serve as triboelectric materials. Natural wood possesses a negligible triboelectric polarizability which limits the ability to generate surface charges. Triboelectric properties are related to the roughness of material. The chosen wood species and the cutting direction can be relevant in the wood-based TENG performance. The surface modification by in situ-grown zeolitic imidazolate framework-8 (ZIF-8) enhances the ability of donor electrons of materials while coating with poly(dimethylsiloxane) (PDMS) enhances the ability of materials to accept electrons. The size of ZIF-8 nanocrystals in ZIF-8@wood material is controlled by the molar ratio between the ligand (2-MeIm) and the metal ions (Zn cations, Zn^2+^). The molar ratio used ranged from 5 to 20, resulting in ZIF-8 particle sizes ranging from 616 nm to 1008 nm. The percentage of ZIF-8 loading in spruce wood cut by cross-section (C) was 11.0 wt%, 9.3 wt%, and 8.8 wt% for molar ratios of 5-, 10-, and 20-ZIF-8@spruce(C) composites, respectively. For molar ratio of 1, ZIF-8 was not formed on spruce(C) composite. The alkaline pretreatment had a dramatic effect on the structure and physical form of the 20-ZIF-8@balsa(C), as it partially removed lignin and hemicelluloses. Morphological characterization demonstrated that the incorporation of ZIF-8 on the surface of wood reduced the surface roughness at the macroscale but increased the roughness at the nanoscale. The increase in molar ratio from 5 to 20 resulted in an increase in nanoscale roughness from 169.1 nm to 286.5 nm. The surface of PDMS@wood is smoother than native wood. PDMS covered the whole surface with 10 um of thickness. The wood microstructure as scaffold allows the creation of nano-roughness via its functionalization with ZIF-8 nanocrystals and PDMS. Some 2 × 3.5 cm^2^ TENGs were fabricated using different molar ratios to control the size of ZIF-8 in wood (balsa, spruce, and yew) with different cutting angles (tangential (T), radial (R), and cross section(C)). The best performance among these TENGs was achieved with the use of a pair of 20-ZIF-8@spruce(R) and PDMS@spruce(R) as triboelectric materials (Figure 9f). It exhibited a Voc of 24.3 V and an Isc of 0.32 µA at 50 N of force applied. It is over 80 times higher than that of a native wood TENG (0.3 V and 0.004 µA). These improvements were attributed to physical changes (surface morphology and an increase of contact surface area) and chemical changes (caused by the introduction of efficient electron-accepting and electron-donating species).

Luo et al. [94] used natural balsa wood to fabricate a wood-based TENG (W-TENG) with improvements in electrical output and mechanical properties. A two-step process was used to treat balsa wood. The first treatment consisted of chemical treatment with NaOH/Na_2_SO_3_ solution, which partially removed the lignin/hemicellulose content. This treatment produced structural changes. The 3D porous structure of wood evolved from latticed cell lumina to a crumpled structure with an irregular shape. Wood triboelectric-based film was obtained after the hot-pressing process. A single-electrode mode TENG was fabricated using this wood-based film and PTFE as triboelectric positive and negative material, respectively. This 3 × 3 cm^2^ W-TENG exhibited a Voc of 81 V, an Isc of 1.8 µA, and a transferred charge density (Δσ) of 36 µC/m^2^ at a frequency of 1 Hz and a pressing force of 20 N. The electrical output of this W-TENG was improved by more than 70% compared to that of natural wood TENG. Under a load of resistance of 40 MΩ, this W-TENG exhibited a maximum peak output power of 51 μW. This TENG had superior stability with a negligible decay in continuous operation of 20,000 cycles. This TENG also has good flexibility, is lightweight (0.19 g), has a thin thickness (0.15 mm), and is cost-effective.

Zhang et al. [95] used recyclable disposable facial towels based on bamboo fiber, wood fiber, and cotton fiber to fabricate TENGs. These towels were washed with anhydrous ethanol, dried, and cut into the size of 3 × 3 cm^2^. The TENGs based on bamboo, wood, and cotton fiber-based films were named PC-, PW-, and PB-TENG, respectively. Fiber based film and PDMS film were used as triboelectric positive and negative materials in the fabrication of TENGs, respectively. Copper films were also used as electrodes, with acrylic serving as the substrate. The lowest TENG performance was achieved by PW-TENG with a Voc of 69.8 V and Isc of 2 μA. The best output performance was achieved by PB-TENG with a Voc of 101.2 V and Isc of 2.65 μA. This TENG exhibited an output power density of 20.56 µ W/cm^2^ with an external load of 50 MΩ. The effect of contact frequency, distance between triboelectric materials, and the area of contact were also studied. In all cases the electrical output increased when the other parameters were fixed.

Li et al. [96] used a cyclic-spraying method to grow surface-attached metal–organic frameworks (SURMOFs) on living leaves to be used as triboelectric material. H_2_F_4_bdc = tetrafluoroterephthalic acid, bdc = 1,4-benzenedicarboxylate, NH_2_-bdc = 2-amino-1,4-benzenedicarboxylate were used to grow SURMOFs with different end groups. H_2_F_4_bdc ligands allow the growth of the SURMOF with -F end groups (F-SURMOF). Using the bdc ligand, SURMOF with -COOH end groups (COOH-SURMOF) was grown, while using the NH2-bdc ligand allowed for the growth of SURMOF with -NH2 end groups (NH2-SURMOF). Some TENGs were developed using a SURMOF@Scindapsus leaf and a PDMS film as the positive and negative layers, respectively. F-SURMOF@Scindapsus leaf based TENG exhibited a Vpp and Ipp of 22 V and 2.7 µA, respectively, while COOH-SURMOF@Scindapsus leaf based TENG and NH2-SURMOF leaf based TENG exhibited a similar electrical outputs Vpp and Ipp of 24 V and 2.8 µA.

## 5. Applications of Lignocellulose-Based TENGs

Shi et al. [104] used lignocellulosic-based triboelectric nanogenerators (AL-TENGs) to develop a self-powered smart ward system and a self-powered contactless medical monitoring system. This was done to reduce the risk of mutual infection and improve patient comfort. These systems utilize human motion to control electrical ward devices and monitor the patient’s condition. The AL-TENG sensor array was fabricated with 8 units of 2 × 2 cm^2^ AL-TENGs. It is designed to generate an electrical signal when the patient touches the panel with their finger. This electrical signal was acquired, processed, and transmitted to monitoring equipment via a circuit controller, a relay, and a wireless transmitter. The patient could easily control the operation of the lamp, curtains, and air conditioner via the panel. Different patients’ symptoms or health conditions are also collected via this panel. These were displayed in real-time on a program interface, and emergency calls or messages could be sent to the physician’s phone when the patient needed prompt medical assistance.

Zhou et al. [87] utilized cellulose-based TENGs in the creation of a floor monitoring system, aiming to facilitate smart homes and the Internet of Things (IoT). This system could collect sensory information from our daily activities, such as position, state (walking, jumping, tumbling, falling), and speed. To collect this information, cellulose-based TENGs were constructed and placed in the floor using single-electrode mode. The electrical signal generated by TENGs was acquired and analyzed using a microcontroller and a signal processing module. This sensor was tested by varying frequencies and pressures to simulate different walking speeds and individuals of varying weights. Pressures in the range of 70 to 300 kPa were applied to a cellulose-based TENG at a fixed frequency of 5 Hz to obtain the output voltage-pressure relationship. These voltages ranged from 300 to 450 V, and there was a linear relationship between the output voltage and pressure with a slope of 0.6 V/kPa. In the same manner, frequencies ranging from 1 to 9 Hz were tested at a constant pressure of 125 kPa. The output voltage ranged from 100 to 500 V, with a correlation coefficient of 0.996, fitting a non-linear curve.

Zhang et al. [88] constructed a humidity environment sensor made from a composite of bamboo cellulose and Ti_3_C_2_Tx, based TENG. The 3D structure of the lignocellulosic matrix after delignification facilitates the diffusion of water molecules and enables the binding of Ti_3_C_2_Tx nanosheets within the matrix. The water contact angle measurement showed an enhancement in hydrophilicity, decreasing from 83.7° for wood to 12.5° in the cellulose/Ti_3_C_2_Tx composite structure. The fabricated TENG was composed of a cellulose/Ti_3_C_2_Tx composite and FEP films as triboelectric positive and negative materials, respectively. The electrical signal generated by this TENG was acquired using a digital multimeter. This multimeter, equipped with a Bluetooth module, transmits data in real time to a mobile phone. This TENG was capable of accurately measuring the relative humidity (RH) of the environment in the range of 40–90% with high sensitivity (0.8/1%) and good performance (150 s). The output voltage decreased from 5 to 3.8 V as the RH increased from 40 to 80%. This system was also capable of measuring the humidity of human breath and finger humidity, showing a linear correlation between output voltages and RH measurements.

Dudem et al. [89] used recycled material-based TENG to develop information and communication systems. A smart wristband was fabricated using a 30 × 30 mm^2^ TENG based on carbon-coated paper wipes (C@WP-TENGs), which served as a Morse code generator. This device could send messages using Morse code by varying the time of contact and separation between triboelectric layers. This C@WP-TENGs was well packed in polythene to protect it from moisture. It also maintained a humid environment (with relative humidity ranging from 45% to 80%) in order to generate a consistent electrical output. The TENG was connected to an electrometer, which was then connected to a computer. In the computer, a customized program decodes the electrical signals and later stores and sends an email notification to a mobile phone with the word “decoded”. Dudem et al. [89] also fabricated a 9-segment keypad composed of 9 C@WP-TENGs. This keypad was designed to send information (letters and numbers) to a computer. It was displayed on the computer screen using the Arduino controller. It was proposed to be useful for self-powered data transmission and as a door lock security system.

Zhang et al. [91] used a stator/rotator system TENG fabricated with PTFE/PSP films as triboelectric negative and positive materials to harvest wind energy. The movement of the rotor was generated via wind because it was connected to cups that collect airflow. The output performance depended on the wind speed. The PSP-based TENG at speeds of 3.9, 6.3, and 9.8 m/s exhibited an Isc of up to 20, 30, and 40 μA, respectively, and a Voc of 40, 80, and 200 V, respectively. It was capable of lighting up approximately 228 commercial LEDs after rectification. This system was also used as a power source to achieve cathodic protection for steel. For this purpose, a Q235 carbon steel sample was polished using 7000 mesh sandpaper and immersed in a 3.5% NaCl solution to simulate seawater. The electric energy generated by the PSP-TENG was injected into the surface of carbon steel to reduce electron migration and prevent corrosion. This PSP-TENG was proposed to be used as an anti-corrosion system for marine pipelines and ships.

Sun et al. [68] used a functionalized wood-based TENG (FW-TENG) array to fabricate a proof-of-a-concept of a smart home. The 10 × 8 cm^2^ FW-TENG was composed of PDMS coated and in situ growth ZIF-8 in spruce veneer wood as triboelectric positive and negative materials. It exhibited Voc and Isc of 79.6 V and 0.94 mA under a force of 50 N, and a maximum instantaneous power of 7.3 mW with the load resistance of 80 MΩ. FW-TENG array was assembled by 6 FW-TENGs connected. It was covered with a larger veneer made of native wood, measuring 35 × 20 × 0.1 cm^3^. This system could easily power a household lamp (2W, E14) when directly connected, as it harnesses the motion of a human adult walking. It also powered an electrochromic window that was initially opaque. The FW-TENG becomes transparent when pressed with a hand. This FW-TENG array demonstrates its applicability in smart buildings.

Luo et al. [94] used a balsa wood-based TENG (W-TENG) to fabricate a smart ping-pong table capable of tracking the position of the ball with high accuracy. They fabricated a proof-of-concept demonstration of this smart ping-pong table with an 8 × 8 array of W-TENGs. Each W-TENG had a size of 4.5 × 4.5 cm^2^, and the gap between two adjacent TENG units was 0.5 cm. This system was connected to a synchronous data acquisition card with integrated signal conditioning, which was used for multi-channel measurements. The acquired data was then transferred to a computer for processing. It enabled real-time sensing via statistical analysis of the position and velocity of the ball. With a higher velocity, the output increased. In the experiments, it was be divided into two regions. In the low-velocity regions (<4.5 m/s) this system obtained a high sensitivity of 0.78 V/ms^−1^ (R^2^ = 0.997) while with velocities beyond 4.5 m/s this system obtained a sensitivity of 0.21 V/m^s−1^ (R^2^ = 0.967). These results could have great value for judging matches because they could provide real-time assistance to the judges. It could also be helpful as a training guide for athletes and referees. It enables smart sport monitoring and assistance.

Shi et al. [105] fabricated natural wood-based TENGs (W-TENGs) to serve as self-powered sensors (WTSS). This TENG was composed of a wood-based film and PTFE film as triboelectric positive and negative materials. The wood was converted into flexible wood by a two-step process which involves chemical boiling in a mixed NaOH/Na_2_SO_3_ solution followed by a hot-pressing process. This 2 × 2 cm^2^ TENG exhibited a Voc of 38 V and an Isc of 0.37 μA under 2.5 kPa. Four W-TENGs was assembled in a 4 channel WTSS array to develop a self-powered music-player control system. This WTSS array was connected to a data-processing microcontroller, a wireless transmitter, and a receiver. This system was capable of playing, pausing, skipping to the next song, and adjusting the volume of music in a computer-installed music player. In a second application, a 3 × 3 WTSS array was used as a number-based door lock system. WTSS systems in both cases harvested human finger touches to generate an output voltage signal and produce the expected response. In order to create a smart floor monitoring system, an 8 × 8 WTSS array with a unit size of 8 × 8 cm^2^ was developed. This system was capable of monitoring the location of a person and also detecting if the person experienced a fainting episode. These systems allow the creation of smart home control system.

Cai et al. [106] fabricated a porous wood-based TENG which was used in the development of wireless gas sensor system (TWGSS). TWGSS was capable to detect ammonia (NH3) concentration which is one of the most used key food spoilage markers gas. Wood was chemically treated to remove some lignin/hemicellulose and later be soaked in a solution of carbon nanotubes (CNTs) to improve the conductive and ammonia sensing properties. This study utilized conductive wood with ammonia sensing properties as the triboelectric positive material, and FEP film as the triboelectric negative material. This 4 × 4 cm^2^ wood-based TENG, operating in vertical contact–separation mode, demonstrated an electrical response that varied with the concentration of ammonia. With a zero concentration of ammonia, this TENG exhibited a Voc of 47 V and Isc of 2.4 µA at a frequency of 2 Hz. The voltage of this TENG decreases to 39, 34.2, 25.5, 11.8, and 5.7 V at ammonia concentrations of 50, 100, 200, 300, 400, and 500 ppm, respectively. This TENG had high robustness because it exhibited a non-significant decrease in electrical output under high humidity (75%), low temperature (−18 °C), or after 20,000 contact–separation cycles.

Xu et al. [107] fabricated a transparent multi-layer cellulose-based film to be used as a triboelectric and electrode material in the development of TENGs. Ethyl cellulose (EC) was deposited onto the surface of a glass substrate, and then silver nanowires (AgNWs) were sprayed onto the surface of this composite layer. A composite layer consisting of EC and triethyl citrate (TEC) was ultimately applied to the AgNW layer. This cellulose-based film (EC-based film) and PDMS film were used as triboelectric positive and negative materials, respectively. A cable was connected to the Ag layer to collect the generated electrical energy. This EC based TENG exhibited a maximum Voc of 150 V and an Isc of 0.74 µA. An arrangement of four single-electrode mode EC-based TENGs was used in the development of a password recognition system. This EC TENG array was connected to a signal detector and then to an Arduino board to collect and process the signals. The composite EC film and the nitrile rubber glove (from the hand glove) were used as positive and negative triboelectric materials, respectively. This system was capable of identifying a previously set password by lighting up an LED with the correct password.

Hao et al. [108] fabricated a natural wood-based TENG (W-TENG) to be used in the development of smart home and smart floor system. Natural wood extracted from New Zealand pine and PTFE film were used as positive and negative triboelectric materials, respectively. This 8 × 8 cm^2^ TENG was based on single-electrode mode and exhibited a maximum voltage of 220 V and maximum current peak value of 5.8 µA at 2 Hz of frequency. This TENG exhibited excellent stability without any electrical drop over 500 contact–separation cycles. A 5 × 5 cm^2^ W-TENG was placed in the door of a mini-house model to light up commercial LEDs. This W-TENG was connected to a signal transmitter/receiver in order to activate the lights and an alarm. This experiment was successfully conducted, demonstrating its capability to develop smart homes. Five 15 × 20 cm^2^ W-TENGs were placed on the floor to develop a smart floor system capable of determining the position of a person. It was proposed to use it for tracking and recording the dancers’ movements in a studio with a wooden floor.

Sun et al. [109] used plasma treatments to enhance the triboelectric properties of wood. O_2_ plasma treatments and C_4_F_8_ + O_2_ plasma treatments improved the triboelectric positive and negative properties, respectively. C_4_F_8_ + O_2_ plasma wood becomes 156 times more tribonegative than native wood. A 20 × 35 × 1 mm^3^ wood-based TENG was fabricated in vertical contact–separation mode and it exhibited a maximum voltage of 42.3 V and a current of 0.6 µA under 40 N of external force. This TENG was also placed on the floor of a mini-house model to harvest energy from human steps (simulated with hand pressing) in order to power small electronic devices such as a lamp, calculator, and a smart window. In the configuration of single electrode mode, the C_4_F_8_ + O_2_ plasma treated wood was used as tribopositive material. This was placed on the floor and some material was tested as tribonegative material. A human wearing silk socks, which served as a tribonegative material, was placed onto this material, and the measured electrical output was −50 V. This TENG, measuring 12 × 10 cm^2^, was capable of powering on an LCD screen. It was proposed to be used as a smart floor system to harvest human steps for use in household devices.

Bi et al. [110] fabricated a porous wood-based TENG (WTENG), which served as a flexible triboelectric sensor (PWFTS). This sliding mode TENG was composed of ring components (stator and rotator) and used to harvest motion energy from human legs and arms. Chemical treatment with sodium hydroxide (NaOH), sodium sulfite (Na_2_SO_3_), and sodium chlorite (NaClO_2_) eliminated the lignin content and partially degraded the hemicellulose within the wood, resulting in the formation of a porous structure. This treatment enhanced the porosity, resulting in increased surface point contacts and improved mechanical properties, such as flexibility. This porous wood was bent at a 180° angle to fabricate a stator–rotator TENG. Porous wood film and PTFE were used as triboelectric positive and negative materials, respectively while PI and Al foil were used as substrate and electrode. PWFTS composed of three WTENGs can be divided into three components: the stator, rotor, and brush. The stator was installed on the body and acted as a cathode/anode to collect electrons. The rotor was connected to the moving section of the body, and the brush made it possible to collect the electrical energy generated. This WTENG exhibited a Voc of 46.7 V and an Isc of 1.5 μA, while a TENG based on natural wood exhibited 18.6 V and 1.5 uA, respectively. The electrical output did not change significantly in voltage when the frequency was between 1 and 4 Hz, and when it was continuously working for up to 160 s. However, the signal was clearly improved when the triboelectric layers were bent. PWFTS was capable of capturing human movements from the arms and legs during typical human activities. Peak-to-peak voltages of 50 V, 100 V, and 60 V were harvested from jumping, walking, and running activities, respectively. This device also was tested over 30 days and in different environmental conditions (RH: 0, 30, 60, and 90%) with no changes in electrical output. The degradation speeds of natural, multi-punched, and porous woods were also tested in the presence of a 10% mass fraction sodium hypochlorite solution. Porous wood was completely degraded within 48 h, while the other types of wood were only partially degraded.

Xiong et al. [111] fabricated a composite material which served as triboelectric material and electrode at the same time. Silver nanowires (AgNWs) were sprayed onto a PFA film to form a mesh of bundled silver nanowires (AgBM), which was then coated with a layer of ethyl cellulose (EC). The EC paper was peeled and the AgBM was transferred. A layer of transparent hydrophobic cellulose oleoyl ester (HCOE) nanocoating was applied to both sides to protect the conductive path and the EC layer. This composite layer served as electrode only in the conductive side of EC paper and as triboelectric material in the other side because the dielectric layer. Single-electrode mode TENGs were fabricated to harvest human, wind, and water mechanical energy. Skin and water droplets acted as triboelectric positive material. PFA film was used as the triboelectric negative material in a TENG to harvest wind energy. A 10 × 10 cm^2^ cellulose electronic paper (40 µm in thickness)-based TENG (CEP-TENG) was used to harvest hand-tapping. This TENG exhibited a Voc of 120 V and a Isc of 1.2 μA/cm^2^. A 4 × 4 cm^2^ CEP-TENG film was used to harvest wind energy and it exhibited a Voc of 10 V and 0.2 μA/cm^2^. A 7 × 14 cm^2^ CEP-TENG was used to harvest flowing water. Output voltages and current densities were correlated to the flowing velocities. Under velocities of 6.2, 7.4, 8.6, and 10.4 m/s, this TENG exhibited a Voc and Isc of 20, 38, 45, and 50 V, and 0.25, 0.4, 0.6, and 0.65 μA/cm^2^, respectively. This TENG exhibited good stability, maintaining a consistent electrical output over a continuous 12-h period.

## 6. Conclusions

The growing use of electronic devices demands the development of new energy sources and storage systems to replace conventional batteries, which are toxic and non-biodegradable. By tapping into the renewable, biocompatible, and biodegradable properties of lignocellulose products, researchers have explored the use of lignocellulose-based triboelectric nanogenerators (LC-TENGs) to harvest mechanical energy from the environment and generate environmentally-friendly electrical energy. Many different techniques involving chemical, thermochemical, and biological processes have been used to isolate the main components of lignocellulose, namely cellulose, lignin, and hemicellulose. These components are mainly used as triboelectric active materials. The electrical performance of LC-TENGs is similar to that reported for TENGs made from synthetic polymers. The generated voltage can reach high values (3 V–700 V), but it is accompanied by low current values (4.5 pA–95 mA). The specific power density of the LC-TENGs reported here ranges from 0.19 mW/m^2^ to 35,100 mW/m^2^. These values are similar to those reported for TENGs made from synthetic and other biomaterials, such as Polyvinylidene Fluoride (PVDF), poly(trifluoroethylene) (PTrFE), chitosan, gelatin, alginate, among others. The highest power density was achieved by LC-TENGs made from half-cell allium plant skins, measuring 35,100 mW/m^2^. The electrical performance not only depends on the materials used as triboelectric active surfaces but can also be enhanced by increasing the superficial contact between the active surfaces. This can be achieved via treatments that modify their micro and nanostructures and increase the quantity of hydroxyl polar groups. These groups provide lignocellulosic materials with an inherent ability to donate electrons. The preparation of lignocellulosic-based composites reinforced with nanomaterials, such as carbon nanoparticles, nanofibers, and nanocrystals, has also been used to improve the electrical conductivity of LC-TENGs.

Researchers have developed LC-TENGs with low environmental impact using industrial, agricultural, and urban lignocellulosic waste. These LC-TENGs, with their inherent biocompatibility, biodegradability, and renewability, offer new possibilities for the development of self-powered devices. These applications encompass various fields, such as health monitoring devices, smart homes and buildings, smart tables, sensor systems, anticorrosion systems, and more. Other researchers are harnessing the dual polarity of living leaves to capture wind energy. There is no doubt that the applications of lignocellulose-based TENGs will continue to increase in the next few years, especially due to the advantage of the resource’s availability. Additionally, the methods for isolation or other treatments will be improved to reduce the environmental impact during their fabrication.

## Figures and Tables

**Figure 1 ijms-24-15784-f001:**
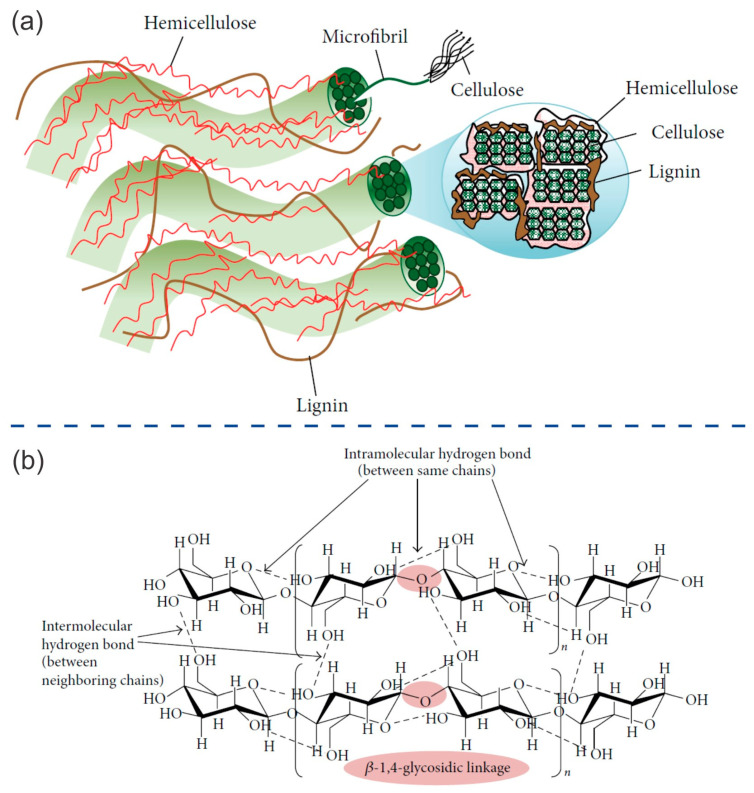
(**a**) Plant structure and microfibril cross-section. Cellulose fibrils are embedded in a matrix of hemicellulose and lignin. (**b**) Cellulose chains and their interactions (Reprinted with permission from Ref. [6]. Copyright 2014 Hindawi).

**Figure 2 ijms-24-15784-f002:**
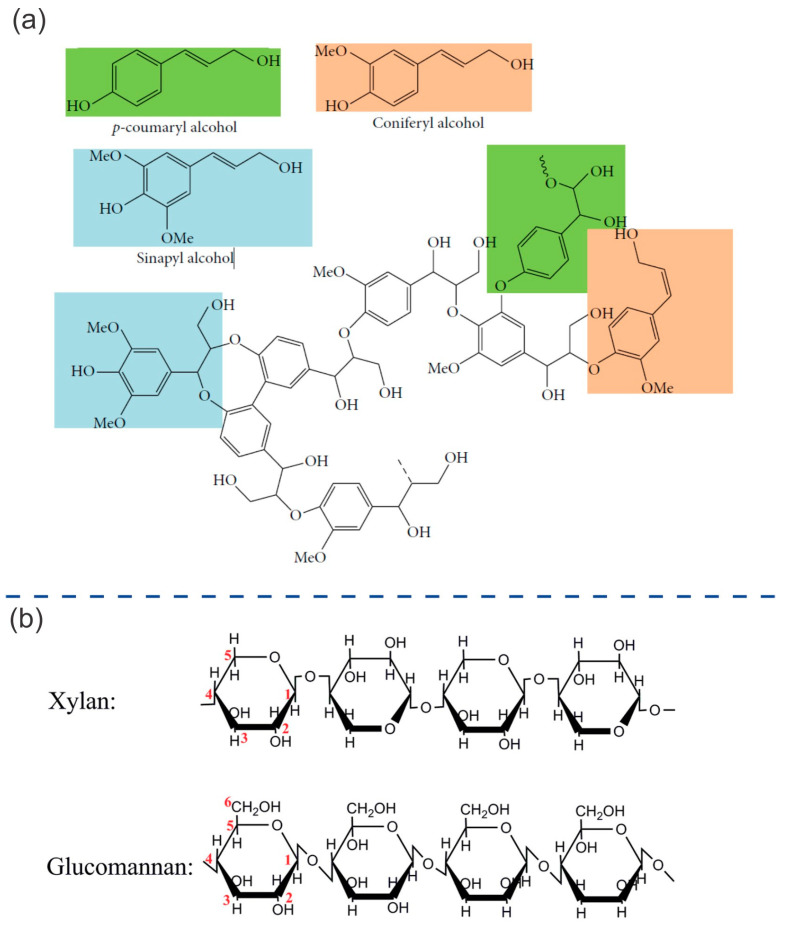
(**a**) Chemical structures of lignin (Reprinted with permission from Ref. [6]. Copyright 2014 Hindawi. (**b**) Chemical structures of hemicellulose (Reprinted with permission from Ref. [26]. Copyright 2011 American Chemical Society).

**Figure 3 ijms-24-15784-f003:**
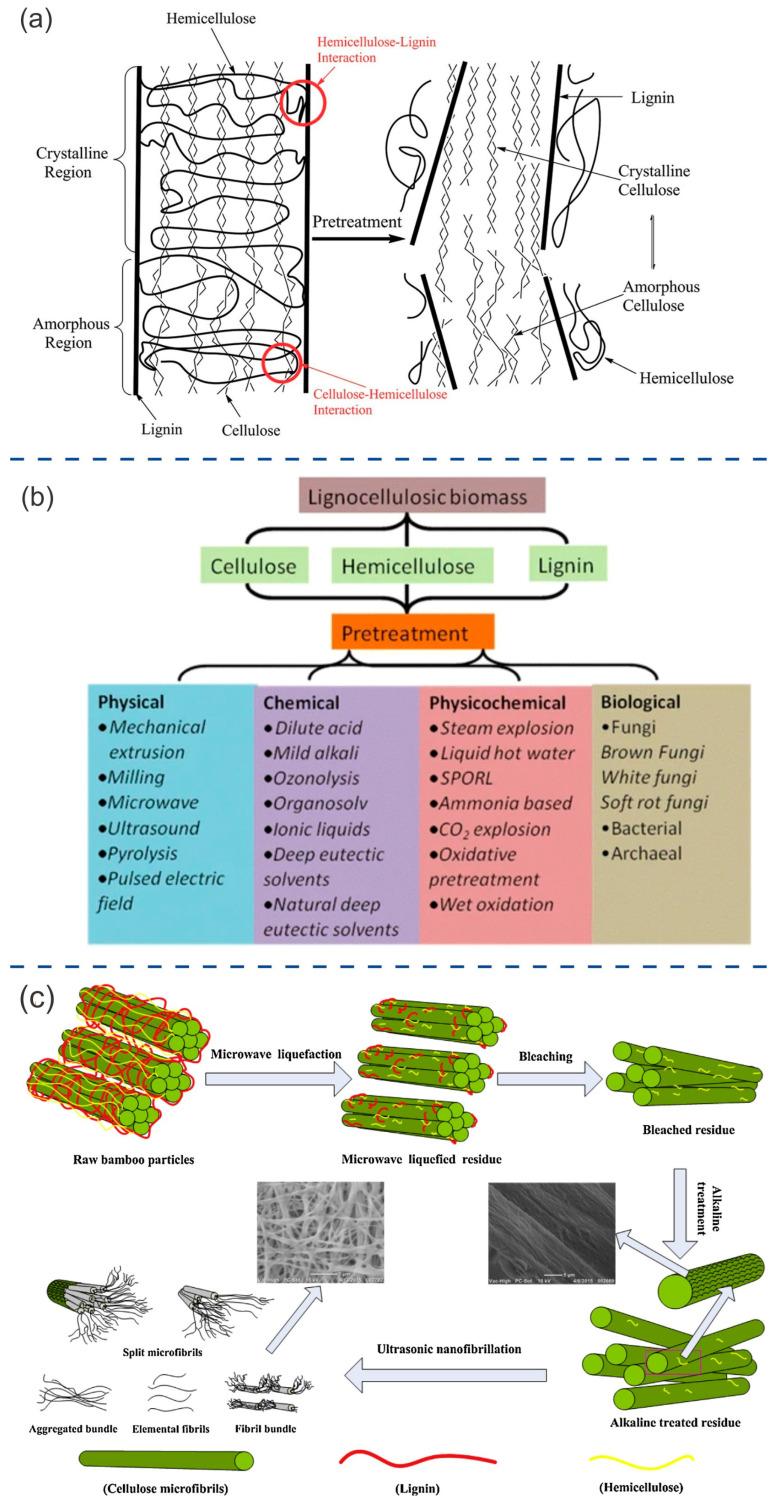
(**a**) Schematic of lignocellulose structure and the effect of pretreatment (Reprinted with permission from Ref. [26]. Copyright 2011 American Chemical Society). (**b**) Traditional pretreatment processes of lignocellulose (Reprinted with permission from Ref. [28]. Copyright 2017 Springer Nature). (**c**) Combination of microwave liquefaction, chemical, and ultrasonication processes to transform bamboo into cellulose microfibrils. (Reprinted with permission from Ref. [29]. Copyright 2016 Elsevier).

**Figure 4 ijms-24-15784-f004:**
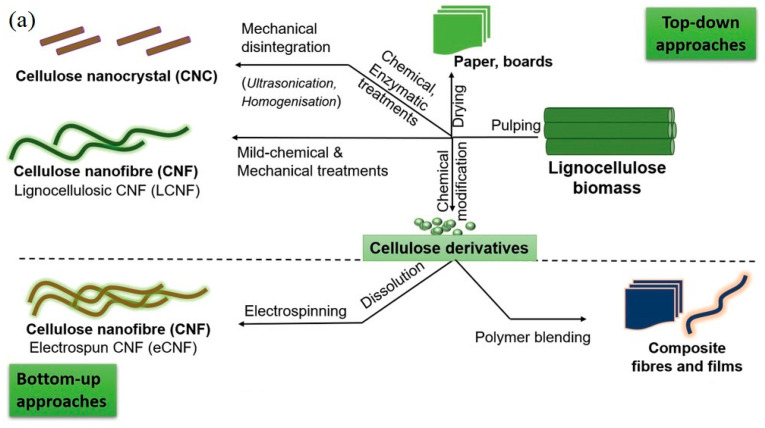
(**a**) Mechanisms for obtaining different types of cellulose-based materials. Top-down approaches start with lignocellulosic biomass, while bottom-up approaches begin with cellulose derivatives to obtain nanomaterials, composite films, and fibers. (Reprinted with permission from Ref. [1]. Copyright 2021 John Wiley and Sons). (**b**) Applications of nanocellulose (Reprinted with permission from Ref. [42]. Copyright 2020 Frontiers in Chemistry).

**Figure 5 ijms-24-15784-f005:**
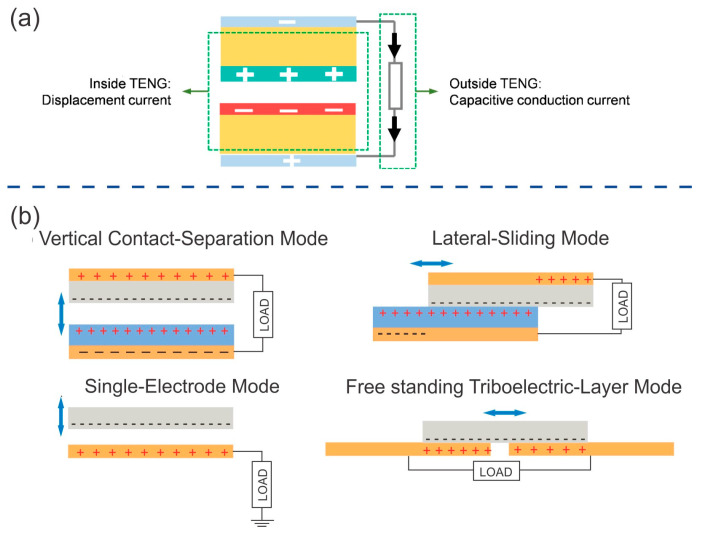
(**a**) Schematic representation of the working mechanism of a wood-based TENG (Reprinted with permission from Ref. [68]. Copyright 2021 Elsevier). (**b**) Applications of nanocellulose (Reprinted with permission [69]. Copyright 2018 John Wiley and Sons).

**Figure 6 ijms-24-15784-f006:**
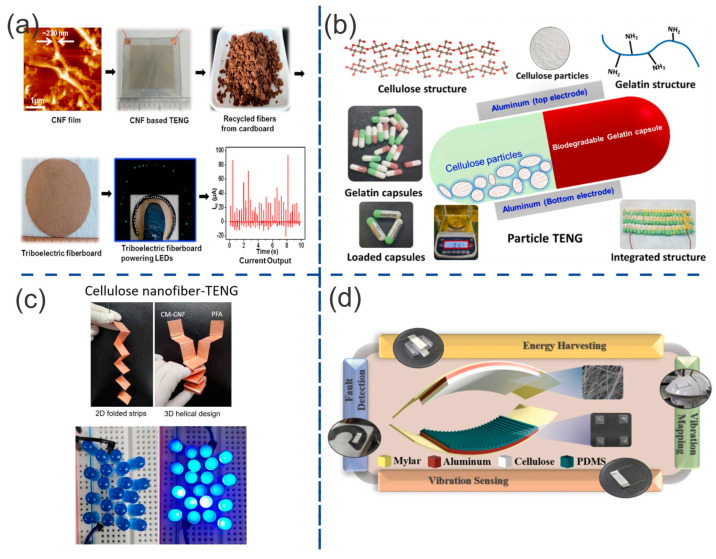
Cellulose-based TENGs: (**a**) cellulose nanofiber-based TENG to harvest human steps in a smart industrial floor (Reprinted with permission from Ref. [71]. Copyright 2016 Elsevier), (**b**) particle TENG (P-TENG) based on cellulose and gelatin (Reprinted with permission from Ref. [67]. Copyright 2022 Elsevier), (**c**) Three-dimensional carboxymethylated cellulose nanofiber TENG (3D CM-CNF TENG). (Reprinted with permission from Ref. [76], Copyright 2022 Elsevier), (**d**) TENG based on electrospun cellulose acetate nanofibers and surface modified PDMS for powering commercial sensors (Reprinted with permission from Ref. [78], Copyright 2022 Elsevier).

**Figure 7 ijms-24-15784-f007:**
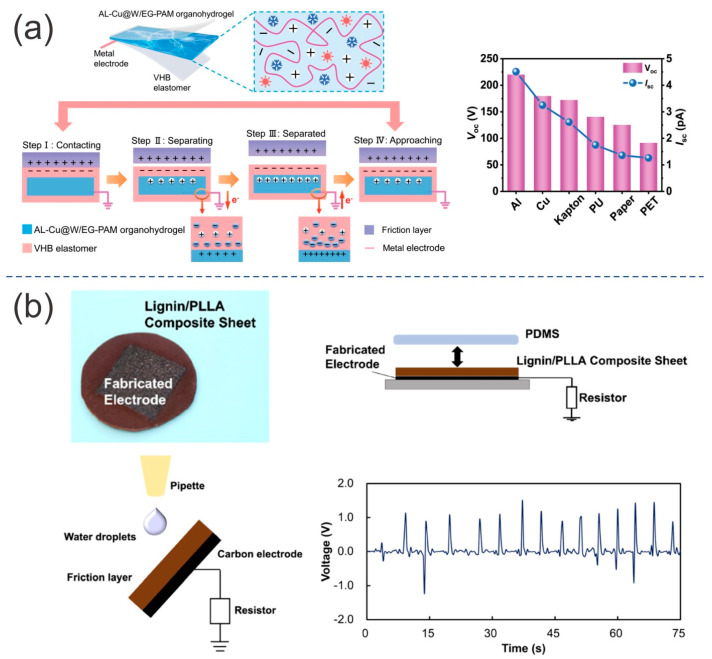
Lignin-based TENGs: (**a**) Lignin-based organohydrogel acting as a triboelectric material and electrode in a TENG (Reprinted with permission from Ref. [81]. Copyright 2022 John Wiley and Sons), (**b**) Lignin/PLLA-based TENG with laser-induced graphitization onto the surface of a lignin-based film acting as an electrode (Reprinted with permission from Ref. [82]. Copyright 2023 American Chemical Society).

**Figure 8 ijms-24-15784-f008:**
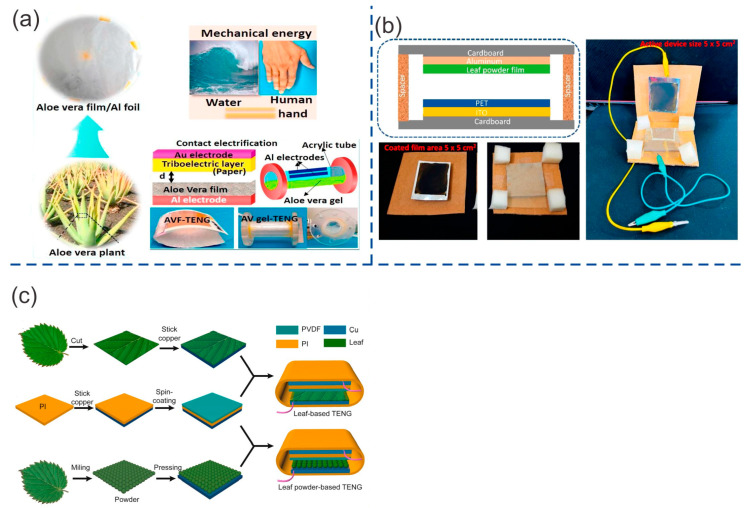
TENGs based on other lignocellulosic components: (**a**) Aloe vera (AV) film and AV gel-based TENG (Reprinted with permission from Ref. [83]. Copyright 2020 Elsevier), (**b**) dry leaf powder-based TENG (Reprinted with permission from Ref. [85]. Copyright 2022 Elsevier), (**c**) Leaf powder and living and dry-based TENGs. (Reprinted with permission from Ref. [86]. Copyright 2019 Elsevier).

**Figure 9 ijms-24-15784-f009:**
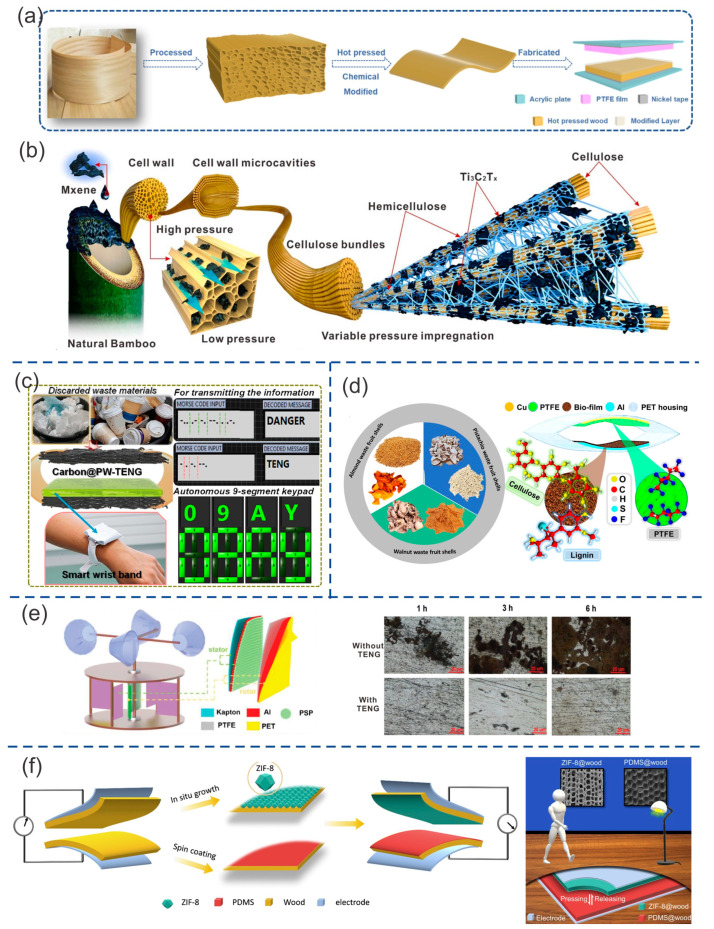
Less and non-refined lignocellulose-based TENGs: (**a**) wood-based TENG (Reprinted with permission from Ref. [87]. Copyright 2022 American Chemical Society), (**b**) Cellulose/Ti_3_C_2_Tx-based TENG with 8 h of delignification time (Reprinted with permission from Ref. [88]. Copyright 2023 Elsevier), (**c**) TENG based on paper wipes (PW) with 15.13 wt% C NPs and PTFE from waste plastic coffee cups. (Reprinted with permission from Ref. [89]. Copyright 2022 American Chemical Society), (**d**) lignocellulosic waste fruit shell based TENG (Reprinted with permission from Ref. [90]. Copyright 2022 Elsevier), (**e**) Rotator-stator system peanut shell powder (PSP) based TENG (Reprinted with permission from Ref. [91]. Copyright 2022 John Wiley and Sons), (**f**) TENG based on 20-ZIF-8@spruce(R) and PDMS@spruce(R) films (Reprinted with permission from Ref. [68]. Copyright 2021 Elsevier).

**Table 1 ijms-24-15784-t001:** TENGs based on lignocellulose biomass.

Lignocellulosic Biomass	Triboelectric Materials	Dimensions and Conditions	Electrical Output	Applications	Refs.
Cellulose	CNF: FEP	40 cm^2^	30 V, 0.035 mA, 0.56 mW (1 MΩ)	Industrial smart-floor	[71]
	Cellulose: Gelatin	Pill size (length 20 mm and diameter 7 mm)	15 V, 0.4 × 10^−3^ mA, 5.488 × 10^−3^ mW	Self-powering human motion-sensing	[72]
	Alc-S5-CNF: PET	2 × 2 cm^2^, 12 N, 20 Hz	7.9 V, 0.051 mA, 101.3 mW/m^2^	Self-powering flexible small electronics	[73]
	CNF/TMS/Si NPs: FEP	5 gears, 3 TENG units	125 V, 0.006 mA, 0.038 mW (5 × 10^7^ Ω)	Harvest water wave energy	[74]
	PEO/CCP-4: PDMS	3 × 3 cm^2^, 40 N, 3 Hz	222.1 V, 0.0043 mA, 217.3 mW/m^2^ (60 MΩ)	Human health detection device	[75]
	CM-CNF210/PFA	100 cm^2^, 10 N, 2 Hz	125 V, 0.012 mA, 2.07 mW (10 MΩ)		[76]
	NR-CF@Ag_3: PTFE	4 × 4 cm^2^, 5 N, 5 Hz	128 V, 0.0124 mA, 3650 mW/m^2^ (0.7 MΩ)		[77]
	CE NF: PDMS	2 × 1 cm^2^, 3 N, 4 Hz	400 V, 3 mA/m^2^, 900 mW/m^2^ (10^8^ Ω)	Self-powered vibration analysis system	[78]
Lignin	0.5 M NaOH 6% glycerol 1:9 lignin-starch: Kapton	6.5 × 6.5 cm^2^, 2.08 N, 0.5 Hz	3.5 V/cm^2^, 23 nA/cm^2^, 1.735 mW/m^2^		[79]
	Lignin NF: PI	21.5 N, 10 Hz	4.5 V		[80]
	Alkali Lignin—Cu@W/EG-PAM: Al	2 × 3 cm^2^, 3 Hz	220 V, 4.5 pA	Monitoring finger and wrist-bending device	[81]
	Lignin/PLLA film: PDMS	13 × 13 mm^2^, 10 N, 1 Hz	15 V, 1.98 mW/m^2^ (200 MΩ)	Environmental harvesting devices	[82]
Other lignocellulosic materials	Aloe Vera (AV) film: PDMS	2.7 cm^2^, 2 m/s^2^	32 V, 0.11 µA, 1.9 mW/m^2^ (100 MΩ)		[83]
	AV gel: PDMS	2.7 cm^2^	12.72 V, 113.23 nA		[83]
	Half-cell Leek leaf skins		182 V, 0.83 mA/m^2^, 35,100 mW/m^2^		[84]
	Leaves powder: PET	5 × 5 cm^2^	3.86 V, 0.0037 mA, 1.894 mW/m^2^ (20 MΩ)		[85]
	leaf powder/PLL: PVDF	4 × 4 cm^2^, 5 Hz	1000 V, 0.06 mA, 17.9 mW (11 MΩ)	Self-powered wind sensor	[86]
	Aloe Vera (AV) film: PDMS	2.7 cm^2^, 2 m/s^2^	32 V, 0.11 µA, 1.9 mW/m^2^ (100 MΩ)		[83]
Non-refined lignocellulosic materials	Lignin/Cellulose/Citric Acid: PTFE	2 × 2 cm^2^, 3.5 kPa, 1 Hz	31 V, 0.0002 mA, 10 mW/m^2^ (80 MΩ)	Smart ward and medical monitoring system	[87]
	paper cardboards	8 × 8 cm^2^	30 V, 0.090 mA	Smart floor system	[71]
	Black walnut with surface modification	3 × 3 cm^2^	335 V, 0.009 mA, 3800 mW/m^2^	Self-powered real motion monitoring device for smart homes	[88]
	Cellulose/Ti_3_C_2_Tx with 8 h of delignification time		60 V, 0.010 mA, 250 mW/m^2^ (7 × 10^5^ Ω)	Self-powered humidity sensor	[89]
	Paper wipes (PW)/15.13 wt% C NPs: PTFE from waste plastic coffee cups	30 × 30 mm^2^, 30 N, 1 Hz	80 V, 0.0015 mA, 0.58 mW (100 MΩ)	Smart wristband and self-powered 9-segment keyboard	[90]
	Pistachio waste fruit shell: PTFE	4.5 × 4.5 cm^2^, 75 N, 10 Hz	700 V, 95 mA, 4161.4 mW/m^2^ (3 MΩ)	Self-powered humidity sensor	[91]
	1000 mesh peanut shell powder (PSP) film: PTFE	2 Hz	225 V, 0.035 mA 365 mW/m^2^ (8 MΩ)	Self-powered wind sensor and metal surface anti-corrosion system	[92]
	F-Wood: C-Wood	2 × 2 cm^2^, 8.2 N, 3 Hz	90.1 V, 0.458 μA, 0.545 mW (47 MΩ)		[93]
	20-ZIF-8@spruce(R):PDMS@spruce(R)	2 × 3.5 cm^2^, 50 N	24.3 V, 0.32 µA	Smart home devices	[68]
	Natural balsa wood film: PTFE	3 × 3 cm^2^, 20 N, 1 Hz	81 V, 1.8 µA, 0.051 mW (40 MΩ)	Smart ping-pong table	[94]
	Cotton fiber-based film: PDMS	3 × 3 cm^2^, 2 Hz	101.2 V, 2.65 μA, 205.6 mW/m^2^ (50 MΩ)	Self-powered wearable sensor	[95]
	COOH-SURMOF@Scindapsus: PDMS NH2-SURMOF: PDMS		24 V, 2.8 µA	Self-powered wind sensor	[96]
Other biomaterial-based TENGs	Gelatin/PLA	4 × 4 cm^2^	5000 mW/m^2^	-	[97]
	Chitosan/frustule silica	3 × 4 cm^2^	15.7 mW/m^2^	Skin-attachable motion sensor	[98]
	Alginate/polyethylene terephthalate	4 × 1 cm^2^	1.33 mW/m^2^	Pressure sensor	[99]
	Natural rubber/polytetrafluoroethylene	4 × 4 cm^2^	237 mW/m^2^	Power source to operate portable electronics	[100]
	Polylactic acid/graphene	2 × 2 cm^2^	5400 mW/m^2^	Biotechnology, optics, and catalyst fields	[101]

## Data Availability

Not applicable.

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
