# Peer review of "Lignocellulosic Biomass for the Fabrication of Triboelectric Nano-Generators (TENGs)—A Review"

_ijms, 2023, doi:10.3390/ijms242115784_

Round 1

Reviewer 1 Report

Review ijms-2579731 "Lignocellulose biomass for the manufacture of triboelectric nanogenerators (TENG) - a review" is devoted to topical, discussed topics. Relevance of dissemination both with the valorization of lignocellulosic biomass and with the growing sector of TENG. The review is written logically, consistently, interestingly and fascinatingly.

It can be improved with minor fixes.

1) Line 49. There is not so much hemicellulose in wheat straw, as the authors believe, only 27-30%. Saini JK, Saini R, Tewari L. Lignocellulosic agricultural waste as a biomass feedstock for second generation bioethanol production: concepts and recent developments // 3 Biotech. - 2015. - No. 5. - P. 337-353 https://doi.org/10.1007/s13205-014-0246-5

2) Lines 195-198 et seq. The authors recommend superior technologies for microwave irradiation and ultrasound, but how can these methods be scaled up to a feedstock that restarts the sequential process of photosynthesis in the amount of ~10x11 tons?

3) Fig. 3 causes the author's error from the source [25], because as a result of the action of ultrasound on the fibrous structure of cellulose, which is characteristic of bacterial nanocellulose, which is a violation of natural physics.

4) The authors of the submission that the isolation of cellulose secretions from lignocellulose is very important, therefore it is necessary to add information about the origin of cellulose in the work (when citing works 68-72, 74), as was done when citing works 67 and 74. It would also be good to indicate the characteristics . detection of cellulose, at least the degree of their purity and dimension, if we are talking about nanocrystalline cellulose and nanostructured cellulose.

5) There is a detailed description for Table 1, but no analysis at all, so the conclusions at the end of the review are not fully justified. Please add an analysis of the properties of TENGs derived from lignocellulosic raw materials and comparable with other raw materials, which are also listed in Table 1.

6) Please note that this review is part of a multi-author review:

Torres, F.G., Gonzalez, K.N., Troncoso, O.P., Korman-Hijar, J.I., & Cornejo, G. (2023). Overview of the development of triboelectric nanogenerators based on biopolymer nanocomposites (Bio-TENG). Applied electronic materials ACS.

What are the features of this review that differ from its size?

7) This is a wish, not a remark. The authors have works:

Torres, F.G., Troncoso, O.P., and De la Torre, G.E. (2022). Hydrogel based triboelectric nanogenerators: properties, performance and applications. International Journal of Energy Studies, 46(5), 5603-5624.

Therefore, I hope that heating elements based on bacterial nanocellulose will soon be obtained. The authors of this section deliberately excluded from this consideration, this is their right.

Author Response

1) The hemicellulose content of wheat straw has been corrected. A new paper has been cited (ln 44-46).

2) Some comments have been added regarding the industrial applications of microwave irradiation and ultrasound (ln 213-216).

3) The references on the Caption of Fig. 3 have been revised. The source of Fig. 3c has been corrected. The description of the effect of ultrasound on the structure of cellulose has been rewritten in order to clarify it.

4) The origin of the cellulose used in works (68-72, 74) was added to the discussion together with other important characteristics such as dimensions, type of cellulose, among others, as long as the works cited reported such characteristics (ln 361, 371, 396-398, 416-420).

5) Two new paragraphs have been added to include a discussion of the electric performance of the TENGs listed in Table 1. The properties of TENGs are analyzed and related to the materials used for the fabrication of the active surfaces (ln 297-339).

6) Some comments regarding a previous review focused on the use of biopolymers for the preparation of TENGs were added to the introduction. The contribution of this paper was compared with the contribution of the previous review (ln. 83-88).

7) No heating elements based on bacterial nanocellulose have been obtained.

Reviewer 2 Report

The authors of the manuscript "Lignocellulosic Biomass for the Fabrication of Triboelectric Nano-Generators (TENGs) – A Review" have undertaken a significant amount of work presenting scientific results on the synthesis and characterization of materials for the production of electrical energy through the triboelectric effect.

I would like to start by mentioning the positive aspects:

1. Numerous material synthesis examples and crucial details for understanding their efficiency are provided.

2. Many instances of triboelectric effect utilization are presented.

3. There are many useful diagrams and figures aiding the comprehension of the presented material.

However, I have some objections that I recommend the authors to consider:

1. Very often, certain sentences are repeated in one form or another in the text. For example, the phrase "Lignocellulosic-based materials are commonly composed of cellulose, lignin, and hemicellulose" is repeated six times just in the first two pages.

2. It would be helpful to explain the difference between a triboelectric nanogenerator and a regular triboelectric generator. In lines 109-110, the following sentence appears: "These nanogenerators utilize bio-based, renewable organic materials as both the active material and layer substrates." This statement doesn't seem to be related to nanogenerators, especially since "nano" primarily refers to dimensions (or energy production capacity?). While the term is commonly used, since you are conducting a review, it would be advisable to provide a more precise explanation.

3. Practical data on the maximum mechanical  and thermal stability of these materials and very few details about their lifespan are presented. If the material is biodegradable (which cannot be said for carbon nanotubes and other additives), it doesn't necessarily take priority, especially if it requires a costly synthesis process for a limited lifespan. Therefore, it would be desirable for the authors to describe the performance of these materials from this perspective.

4. No data about other types of materials are presented; hence, the performance of lignocellulosic materials is not clearly evident.

5. No conclusion is drawn highlighting which materials are currently more efficient in terms of production and/or electric current generation. The conclusion needs to be rephrased.

6. The section describing methods for separating cellulose, lignin, and hemicellulose is valuable but overly extensive.

Author Response

  1. The manuscript has been revised in order to remove repeated sentences and concepts, following the Reviewer #2 suggestion.

  1. A new explanation of how TENGs work, what active surfaces are and why they are called “nanogenerators” has been added to the Introductions (ln 54-74). The sentence “These nanogenerators utilize bio-based, renewable organic materials as both the active material and layer substrates” has been rewritten, following the reviewer #2 suggestion.

  1. A new paragraph has been added to Section 4 in order to discuss data regarding the performance of LC-TENGs from the perspective of environmental impact (ln 321-339).

  1. Table 1 has been modified in order to include the electrical performance of other types of TENGs for comparison. The discussion of the electrical performance of TENGs has been expanded to include the comparison.

  1. The Conclusion section has been rewritten to include discuss which materials are currently more efficient in terms of production and/or electric current generation (1249-1272.

  1. Sections "Extraction of Cellulose and Derivatives", "Extraction of Lignin", and "Extraction of Hemicellulose" have been rewritten and reduced, following the reviewer #2 suggestion.

Reviewer 3 Report

This paper is a review thesis on studies producing 'Triboelectric Nano-Generators' using lignocellulosic biomass. This thesis deals with an interesting topic recently and is well-qualified as a review article referring to many preceding studies.

However, one regrettable point is that there needs to be more information on the production yield of Triboelectric Nano-Generators from previous studies. Yield is critical information in review papers related to producing these products. If the yield of triboelectric materials (CNF, etc.) produced from cellulose, lignin, or other biomass or biomass components is added to Table 1, the thesis will be completed.

Author Response

The production yield was not added to Table 1 as the information was, in most cases, not reported in the papers cited. However, 2 new paragraphs were added to Section 2 to provide general information regarding the production yield of cellulose and lignin (ln 220-232 and ln 244-250).